# EFFICIENT-VDIT: EFFICIENT VIDEO DIFFUSION TRANSFORMERS WITH *Attention Tile*

## ABSTRACT

Despite the promise of synthesizing high-fidelity videos, Diffusion Transformers (DiTs) with 3D full attention suffer from expensive inference due to the complexity of attention computation and numerous sampling steps. For example, the popular Open-Sora-Plan model consumes more than 9 minutes for generating a single video of 29 frames. This paper addresses the inefficiency issue from two aspects: 1) Prune the 3D full attention based on the redundancy within video data; We identify a prevalent *tile-style repetitive pattern* in the 3D attention maps for video data, and advocate a new family of sparse 3D attention that holds a linear complexity w.r.t. the number of video frames. 2) Shorten the sampling process by adopting existing multi-step consistency distillation; We split the entire sampling trajectory into several segments and perform consistency distillation within each one to activate few-step generation capacities. We further devise a three-stage training pipeline to conjoin the low-complexity attention and few-step generation capacities. Notably, with 0.1% pretraining data, we turn the Open-Sora-Plan-1.2 model into an efficient one that is $7.4 \times -7.8 \times$ faster for 29 and 93 frames 720p video generation with a marginal performance trade-off in VBench. In addition, we demonstrate that our approach is amenable to distributed inference, achieving an additional $3.91\times$ speedup when running on 4 GPUs with sequence parallelism.

## 1 INTRODUCTION

Diffusion Transformers (DiTs) based video generators can synthesize long-horizon, high-resolution, and high-fidelity videos (Peebles & Xie, 2023; OpenAI, 2024; Kuaishou, 2024; Lab & etc., 2024; Zheng et al., 2024; Esser et al., 2023; Yang et al., 2024b). The 3D attention is a core module of such models. It flattens both the spatial and temporal axes of the video data into one long sequence for attention computation and reports state-of-the-art generation quality (Lab & etc., 2024; Yang et al., 2024b; Huang et al., 2024). The computation of 3D attention often consumes the majority of the time during the entire forward propagation of a 3D DiT, especially with long sequences when generating extended videos. Thus, existing 3D DiTs suffer from prohibitively slow inference due to the slow attention computation as well as the multi-step diffusion sampling procedure.

This paper tackles the issue by simultaneously sparsifying 3D attention and reducing sampling steps to accelerate 3D DiTs. To explore the redundancies in video data (recall that by nature videos can be efficiently compressed), we examine the attention states in 3D DiTs and identify an intriguing phenomenon, referred to as the *Attention Tile*. As shown in Figure 1(a), the attention maps exhibit uniformly distributed and repetitive *tile blocks*, where each tile block represents the attention between latent frames[1]. This repetitive pattern suggests that *not every latent frame needs to attend to all others*. Moreover, the *Attention Tile* pattern is almost independent of specific input (Figure 1). With these, we propose a solution that replaces the original attention with a fixed set of sparse attention mask during inference (§3.3). Specifically, we constrain each latent frame to only attend to a constant number of other latent frames, reducing the complexity of attention computation from quadratic to linear.

We then consider shortening the sampling process of a video from 3D DiT to further amplify the acceleration effect. Inspired by the recent advance in diffusion distillation (Salimans & Ho, 2022; Song

---

[1] we use the term latent because DiTs compute in the latent space of VAEs (Rombach et al., 2022b).

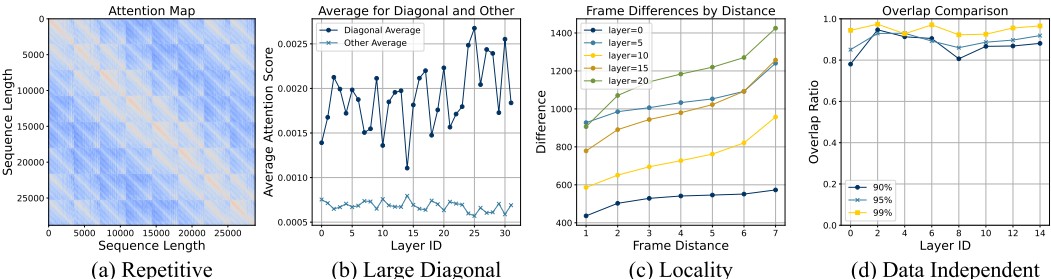

(a) Repetitive      (b) Large Diagonal      (c) Locality      (d) Data Independent

Figure 1: We observe the *Attention Tile* pattern in 3D DiTs. (a) the attention map can be broken down into smaller repetitive blocks. (b) These blocks can be classified into two types, where attention weights on the diagonal blocks are noticeably larger than on off-diagonal ones. (c) These blocks exhibit locality, where the attention score differences between the first frame and later frames gradually increases. (d) The block structure is stable across different data points, but varies across layers. We randomly select 2 prompts (one landscape and one portrait) and record the important positions in the attention map that accounted for 90% (95%, 99%) of the total. We printed the proportion of stable overlap of important positions across layers.

et al., 2023; Kim et al., 2023; Liu et al., 2023b; Sauer et al., 2023; Yin et al., 2024; Heek et al., 2024; Xie et al., 2024), we adopt a simple yet effective multi-step consistency distillation (MCD) (Heek et al., 2024) technique into our approach to achieve the efficient generation of compelling videos. In particular, we split the entire sampling trajectory into adjacent segments and perform consistency distillation within each one. We also progressively decrease the number of segments to improve the generation quality at rare steps.

Due to the orthogonality between sparse attention and MCD, a naive combination is possible, such as directly distilling a sparse student 3D DiT from a pre-trained model. However, the initial gap between the sparse student and the teacher can be large so that the training suffers from a cold start. To tackle this issue, we introduce a more refined model acceleration process named EFFICIENT-VDIT. Initially, MCD is utilized to generate a student model with the same architecture but fewer sampling steps than the teacher. Subsequently, we determine the optimal sparse attention pattern for each head of the student and then apply a knowledge distillation procedure to the sparse model to maintain performance. With 0.1% the pretraining data, we train Open-Sora-Plan-1.2 models into variants that are $7.8\times$ and $7.4\times$ faster, with a marginal performance trade-off in VBench. (Huang et al., 2024). In addition, we provide evidence that our approach is amenable to advances in distributed inference systems, achieving an additional $3.91\times$ speedup when running on 4 GPUs.

In summary, our contribution are:

1. We discover and analyze the phenomenon of *Attention Tile* in 3D full attention DiTs, and propose a family of sparse attention mask with linear complexity to address the redundancy.

2. We design a framework EFFICIENT-VDIT based on our analysis of *Attention Tile*, which turns a pre-trained 3D DiT to a fast variant in a data efficient manner.

3. We evaluate on two Open-Sora-Plan 1.2 models for 29 frames and 93 frames generation. EFFICIENT-VDIT achieves up to $7.8\times$ speedup with little performance trade-off on VBench and CD-FVD. We further demonstrate the potential of integrating our method with advanced distributed inference techniques, achieving additional $3.91\times$ with 4 GPUs.

## 2 RELATED WORK

**Video Diffusion Transformers** There is a rich line of research in diffusion based models for video generation (Ho et al., 2022; He et al., 2022; Luo et al., 2023; Wang et al., 2023c; Ge et al., 2023a; Chen et al., 2024b; Guo et al., 2023; 2024). More recently, Peebles & Xie (2023) introduces the architecture of Diffusion Transformers (DiTs), and several popular video generation models have been developed using the DiTs backbone, for instance, Ma et al. (2024); Zheng et al. (2024); Lab & etc. (2024); Yang et al. (2024b). More specifically, Lab & etc. (2024); Yang et al. (2024b) has explored the use of 3D Full Attention Transformers, which jointly model spatial and temporal

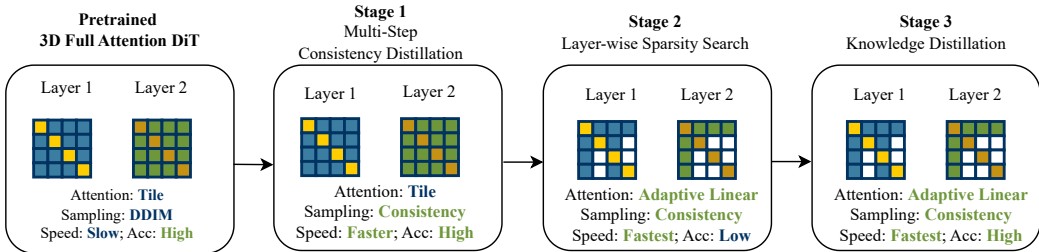

Figure 2: EFFICIENT-VDIT takes in a pre-trained 3D Full Attention video diffusion transformer(DiT), with slow inference speed and high fidelity. It then operates on three stages to greatly accelerate the inference while maintaining the fidelity. In Stage 1, we modify the multi-step consistency distillation framework from (Heek et al., 2024) to the video domain, which turned a DiT model to a CM model with *stable* training. In Stage 2, EFFICIENT-VDIT performs a searching algorithm to find the best sparse attention pattern for each layer. In stage 3, EFFICIENT-VDIT performs a knowledge distillation procedure to optimize the fidelity of the sparse DiT. At the end, EFFICIENT-VDIT outputs a DiT with linear attention, high fidelity and fastest inference speed.

relationship, instead of previous models that separately model spatial and temporal relationship (e.g. one Transformer layer with spatial attention and the other with temporal attention (Zheng et al., 2024; Ma et al., 2024)). The design of 3D full attention has gained increasing popularity due to their promising performance. In this work, we tackle the efficiency problem specifically for 3D full attention diffusion Transformers. In addition, there is a line of research that combines video diffusion model with sequential or autoregressive generation. These methods may also achieve speedup due to their use of shorter sequence length. EFFICIENT-VDIT aims to speedup in a single diffusion forward, which is compatible with orthogonal to autoregressive manner methods (Henschel et al., 2024; Xiang et al., 2024; Chen et al., 2024a; Valevski et al., 2024).

**Accelerating diffusion inference** Many work in diffusion models have been proposed to reduce the number of sampling steps to accelerate diffusion inference (Song et al., 2020; Lu et al., 2022a;b) (Liu et al., 2024). Song et al. (2023) proposes the consistency models which distills multiple steps ODE to one step. Wang et al. (2023b) extends CMs to video generation model. Li et al. (2024b) further extends the idea with reward model to speed up video diffusion model inference. Another line of research that accelerates diffusion models inference utilize multiple devices (Li et al., 2024c; Wang et al., 2024a; Chen et al., 2024d; Zhao et al., 2024). These works exploit the redundancy between denoising steps and use stale activations in distributed inference to hide communication overhead, and are naturally incompatible with work that reduce the redundancy between steps. In this work, we exploit the redundancy in attention computation, which is orthogonal to works that leverage distributed acceleration and redundancy between denoising steps. Our pipeline integrates a multi-step CM approach (Xie et al., 2024) by default, and in experiment, we show that it can also seaminglessly integrate with parallel inference.

**Sparsity in Transformer inference** has been investigated in the context of Large Language Models (LLMs) inference, which can be decomposed into pre-filling and decoding stages (Yu et al., 2022). StreamingLLM discovers the pattern of Attention Sink, and keeps a combination of first few tokens and recent decoded tokens during decoding phrase (Xiao et al., 2023). Zhang et al. (2024a;b) adaptively identify the most significant tokens during test time. Video DiTs have different workload than LLMs, where DiTs perform a single forward in each diffusion step without a decoding phrase. In particular, our paper is among the first to explore sparse attention in the context of 3D Full Attention DiTs. In addition, our finding that *Attention Tile* is data-independent motivates us to design a solution which does not require inference time adaptive searching, which is a bottleneck in work such as Zhang et al. (2024b). Sparsity has also been studied in Gan and other diffusion-based models, yet we focus on the new architecture 3D DiT (Li et al., 2020; 2022). A recent paper (Wang et al., 2024b) also discusses the redundancy in DiTs models, but no performance has been shown.

## 3 EFFICIENT-VDIT

EFFICIENT-VDIT is a framework that takes in a 3D full attention DiT model $T$, and outputs a DiT that runs efficiently during inference $T_{\text{Fast}}$. EFFICIENT-VDIT consists of three stages. The first

stage (§3.2) performs a multi-step consistency distillation and outputs $T_{\text{MCM}}$, following the method developed in image diffusion models (Xie et al., 2024). The second stage (§3.3) takes in $T_{\text{MCM}}$, performs a one-time search to decide the optimal sparse attention mask for each layer, and outputs a model $T_{\text{Sparse}}$ with the optimal sparse attention mask. The last step(§3.4) performs a knowledge distillation to preserve the model performance, using $T_{\text{MCM}}$ as the teacher and $T_{\text{Sparse}}$ as the student, following the distillation design in (Gu et al., 2024; Jiao et al., 2019).

In this section, we first introduce the characteristics of *Attention Tile* that motivate the design of the sparse patterns in Section 3.1. Then, we will introduce the framework EFFICIENT-VDIT by stages.

### 3.1 PRELIMINARY: CHARACTERISTICS OF *Attention Tile*

In §1, we briefly describe that the attention map consists of repetitive tile blocks. In this section, we dive into three characteristics that lead to our design and usage of a family of sparse attention masks.

**Large Diagonals** Tile blocks on the main diagonals has higher attention scores than off-diagonal ones. In Figure 1(b), we plot the attention scores at the main diagonal tile blocks, compared to attention scores at the off-diagonal blocks, on Open-Sora-Plan-1.2 model (Lab & etc., 2024). We find that on average the main diagonal blocks contain values $2.80\times$ higher than the off-diagonal ones. This suggests a separate treatment of tile blocks on and off the main diagonals.

**Locality** Off-diagonal tile blocks are similar, but the similarity decreases with further distance. In Figure 1(c), we plot the relative differences between the first latent frame and subsequent latent frames. We find that the differences increase monotonically. This indicates a need to retain the computation of several tile blocks (i.e. more than one) to accommodate information in distant tile blocks.

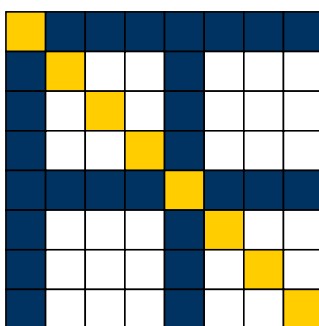

Figure 3: Exemplar attention mask $(2 : 6)$. It maintains the attention in the main diagonals and against 2 global reference latent frames. Tile blocks in white are not computed.

**Data Independent** The structure of the tile is relatively stable across different inputs. We plot the overlap of indices for largest attention scores for different prompts. We observe that roughly 90% of them coincide. This suggests reusing a fixed set of attention masks during inference for different inputs.

Motivated by the above characteristics, we develop a family of sparse attention masks where we keep the attention computation in the main diagonal and the attention with a constant number of global reference latent frames. Figure 3 visualizes one instance of the attention mask. The formulation will be introduce formally in § 3.3.

### 3.2 STAGE 1: MULTI-STEP CONSISTENCY DISTILLATION

We follow (Xie et al., 2024) to perform a multi-step latent consistency distillation(MLCD) procedure to obtain $T_{\text{MCM}}$ as classic CM map from an arbitrary ODE trajectory state to the endpoint. MLCD generalize CM by dividing the entire ODE trajectory in latent space into $S$ segments and carrying out consistency distillation for each segment independently which reduce the difficulty for training dramatically. MLCD obtains a set of milestone states marked as $\{t_{\text{step}}^s\}_{s=0}^{S}$. The loss for MLCD is:

$$\mathcal{L}_{\text{MLCD}} = \left\| \text{DDIM}\left(z_{t_m}, f_\theta(z_{t_m}, t_m), t_m, t_{\text{step}}^s\right) - \text{nograd}\left(\text{DDIM}\left(z_{t_n}, f_\theta(z_{t_n}, t_n), t_n, t_{\text{step}}^s\right)\right) \right\|_2^2$$

where $s$ is uniformly sampled from $\{0, \ldots, S\}$, $t_m$ is uniformly sampled from $[t_{\text{step}}^s, t_{\text{step}}^{s+1}]$, $t_n$ is uniformly sampled from $[t_{\text{step}}^s, t_m]$, $\text{DDIM}(z_{t_m}, f_\theta(z_{t_m}, t_m), t_m, t_{\text{step}}^s)$ means one-step DDIM transformation from state $z_{t_m}$ at timestep $t_m$ to timestep $t_{\text{step}}^s$ with the estimated denoised image $f_\theta(z_{t_m}, t_m)$ and *nograd* refers to one-step diffusion without guidance scale.

---

**Algorithm 1** Searching for the optimal set of sparse attention masks

---

**Require:** Available mask list from dense to sparse $[\text{Mask}_1, \text{Mask}_2, ..., \text{Mask}_n]$, teacher model $M_T$, student model $M$, loss function $\mathcal{L}$, number of timestep samples $m$.
**Require:** Forward function FORWARD, threshold $r$, which is the maximum tolerance for $\mathcal{L}$.
**Require:**
  1: **for** each layer $l$ in model layers **do**
  2:      Initialize best_mask $\leftarrow$ None
  3:      **for** $i$ from 1 to $n$ **do**                ▷ Iterate over masks from dense to sparse
  4:          Apply $\text{Mask}_i$ to the current layer $M^{(l)}$
  5:          Initialize $\mathcal{L}_i^{\max} \leftarrow -\infty$           ▷ Initialize max loss for this mask
  6:          **for** each timestep $t$ sampled $m$ times from $\text{Uniform}(0, 1)$ **do**
  7:              $\hat{y} \leftarrow \text{FORWARD}(M_T^{(l)}, \text{Mask}_i, t)$
  8:              Compute $\mathcal{L}_i(t) \leftarrow \mathcal{L}(y, \hat{y})$
  9:              Update $\mathcal{L}_i^{\max} \leftarrow \max(\mathcal{L}_i^{\max}, \mathcal{L}_i(t))$     ▷ Update the maximum loss
 10:          **end for**
 11:          **if** $\mathcal{L}_i^{\max} < r$ **then**
 12:              best_mask $\leftarrow \text{Mask}_i$      ▷ Update the best mask if max loss is within threshold
 13:          **else**
 14:              **break**
 15:          **end if**
 16:      **end for**
 17:      Assign best_mask to the current layer $M^{(l)}$
 18: **end for**

---

### 3.3 STAGE 2: LAYER-WISE SEARCH FOR OPTIMAL SPARSE ATTENTION MASK

**Sparse Attention Masks** Following our analysis in §3.1, a desired sparse attention mask should separately treat on and off diagonal tile blocks, leverages the repetitive pattern in off-diagonal tile blocks while considering locality. In this paper, we aim on a family of masks that achieve linear compute complexity while prioritizing simplicity and implementation efficiency. Specifically, we simply keep tile blocks in the main diagonals(marked as golden color in Figure 3). For off-diagonal tile blocks, we keep a constant number of $k$ latent frames, and only retain attention between against these "global reference frames" (mark as blue color in Figure 3). Since $k$ is constant, the overall complexity of the attention is linear with respect to the number of latent frames. For simplicity, we choose these $k$ reference frames uniformly from all $F$ latent frames. For clarity, we denote a mask with two numbers - $k : F - k$. For example, the example figure 3 shows an attention mask of $2 : 6$.

**Layer-wise Searching For Attention Masks** Previous studies has suggested that different layers exhibit different amount of sparsity (Wang et al., 2023a; Ge et al., 2023b; Yang et al., 2024a). Using the MSE difference of the final hidden states as a guidance, we develop a searching method to find the best combinations of attention masks across layers (Algorithm 1). Intuitively, we first perform a profiling process on $T_{MCM}$. The profiling step loops over layers, and greedily selects the largest $k$ which does not incur a higher MSE difference than a prede-

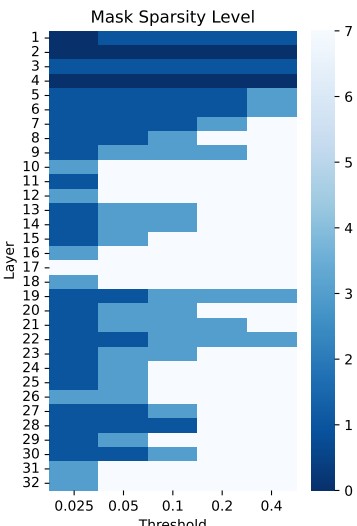

Figure 4: Search results for Open-Sora-Plan v1.2 model (29 frames). We verify that different layers have different sparsity in 3D video DiTs.

fined threshold $r$. A dynamic programming based alternative is also described in Appendix A, where given a runtime constraint, the minimum possible maximum loss difference is computed. In the experiment section ( § 4), we show evidence that this is a key to maintaining video quality. For simplicity, we apply the greedy version of the search throughout the main paper. Figure 3.4 shows an exemplar algorithm output.

### 3.4 Stage 3: Knowledge Distillation with $T_{TCM}$

Stage 2 introduces performance drop since we significantly modify the attention mask. In Stage 3, we apply the method of knowledge distillation, using the model with full attention $T_{MCM}$ as the teacher, and the model with sparse attention $T_{Sparse}$ as the student (Hinton, 2015). We follow a similar design as knowledge distillation methods in Transformer models for Languages (Gu et al., 2024; Jiao et al., 2019), which combines the loss from attention output and hidden states output, over $L$ total layers.

$$\mathcal{L}_{\text{total}} = \frac{1}{L} \left( \sum_{i=1}^{L} \left( \mathcal{L}_{\text{attention}}^{(i)} + \mathcal{L}_{\text{mlp}}^{(i)} \right) \right) + \lambda \mathcal{L}_{\text{diffusion}}, \tag{1}$$

where each term is defined as follows:

**Attention Loss** $\mathcal{L}_{\text{attention}}$: To calculate $\mathcal{L}_{\text{attention}}^{(i)}$, we apply the MSE loss between the output of the student's self-attention layer $\hat{O}_{\text{attn}}^{(i)}$ and the teacher's self-attention layer output $\tilde{O}_{\text{attn}}^{(i)}$:

$$\mathcal{L}_{\text{attention}}^{(i)} = \text{MSE}(\hat{O}_{\text{attn}}^{(i)}, \tilde{O}_{\text{attn}}^{(i)}). \tag{2}$$

**MLP Loss** $\mathcal{L}_{\text{mlp}}$: We calculate $\mathcal{L}_{\text{mlp}}^{(i)}$ as the MSE between the outputs of the student's MLP layer $\hat{O}_{\text{mlp}}^{(i)}$ and the teacher's MLP layer output $\tilde{O}_{\text{mlp}}^{(i)}$:

$$\mathcal{L}_{\text{mlp}}^{(i)} = \text{MSE}(\hat{O}_{\text{mlp}}^{(i)}, \tilde{O}_{\text{mlp}}^{(i)}). \tag{3}$$

In addition, we keep the diffusion loss $\mathcal{L}_{\text{diffusion}}$ for the student model. In practice, we observed that the diffusion loss tends to be an order of magnitude smaller compared to other losses. To balance the contribution of the diffusion loss during the training process, we scale it by a factor $\lambda$, ensuring it has a comparable impact on the overall loss function.

## 4 Experiment

We first present our experiment settings and evaluation metrics in §4.1. We then discuss system performance in §4.2, demonstrating the effectiveness on a single GPU and applicable to multiple GPUs. In §4.3, we compare the video quality with and without variants of our methods with VBench and CD-FVD (Huang et al., 2024; Ge et al., 2024). Finally, we show visualization results in §4.4 of the generation quality for the original model, the MLCD model, and the final model.

### 4.1 Experiment setup

**Models.** We use the 29 and 93 frames models of the popular 3D DiT based Open-Sora-Plan family (Lab & etc., 2024). The model uses VAE inherits weights from the SD2.1 VAE (Rombach et al., 2022a), with a compression ratio of 4x8x8 (temporal, height and width). For the text encoder, it uses mt5-XXL as the language model, and it incorporates RoPE as the positional encoding (Xue, 2020; Su et al., 2024). In addition to the VAE encoder, videos are further processed by a patch embedding layer that downsamples the spatial dimensions by a factor of 2. The videos tokens are finally flattened into a one-dimensional sequence across the frame, width, and height dimensions.

**Metrics.** We evaluate video quality using VBench and Content-Debiased Frechet Video Distance (FVD) (Huang et al., 2024; Ge et al., 2024). VBench assesses the quality of video generation by aligning closely with human perception , computed for each frame of the video and then averaged across all frames, providing a comprehensive assessment. CD-FVD measures the distance between the distributions of generated and real videos toward per-frame quality over temporal realism.

**Baselines.** We consider two models as the major baselines: the original Open-Sora-Plan model and the model after consistency distillation. Following the default settings of Open-Sora-Plan models Lab & etc. (2024), we use 100 DDIM steps for the original model, which is consistent across all experiments and training in the paper. For the MLCD model, we select the checkpoint with 20 inference steps as we empirically find that it achieves the best qualitative result.

**Implementation details.** We use FlexAttention from PyTorch 2.5.0 (Ansel et al., 2024) as the attention backend. We provide a more detailed description on how to leverage FlexAttention to implement our method in Appendix B. We generate videos based on the VBench standard prompt list for VBench evaluation. To avoid potential data contamination in CD-FVD evaluation, we use a set of 2000 samples from the Panda-70M (Chen et al., 2024c) test set to build our real-world data comparison. As we use the CD-FVD score between real-world data and generated videos to evaluate the capacity of DiT models, the prompt style needs to align with the real-world data clip samples. Therefore, we randomly select prompts from the Panda-70M test set caption list for video generation by the models.

**Training details.** All models are trained using the first 2000 samples from the Open-Sora-Plan's mixkit dataset.The global batch size is set to 2, and training is conducted for a total of 10000 steps, equivalent to 10 epochs of dataset. The learning rate is 1e-5, and the gradient accumulation steps is set to 1. The diffusion scale factor $\lambda$ is 100. The MLCD model is trained with 100 DDIM steps of the original model. The final model is trained with a 20-step MLCD model checkpoint.

## 4.2 System Performance

The major target of Efficient-vDiT accelerates inference in a single GPU by using multi-step consistency distillation and sparse attention. In §4.2.1, we demonstrate the system speedup with various settings. In addition, we demonstrate an advantage of our method that it can be seaminglessly integrate with advanced parallel method, i.e. sequence parallelism, in §4.2.2.

### 4.2.1 Efficient-vDiT speedup on a single GPU

We test our approach on a single A100-SXM 80GB GPU. Table 1 shows the computation time for a single sparse attention kernel, while Table 2 presents the average execution time of all layers after layerwise search in Algorithm 1. '2:6' refers to 2 global reference frames in Figure 3. Sparsity refers to the proportion of elements in the kernel that can be skipped. During testing, we consider only the attention operation, where the inputs are query, key, value, and mask, and the output is the attention output. We do not account for the time of VAE, T5, or embedding layers. The measurement method involves 25 warmup iterations, followed by 100 runs. The median of the 20th to 80th percentile performance is used as the final result.

In Table 1, we observe that as the sparsity increases, the computation time decreases significantly. For instance, with a 2:6 attention mask, corresponding to a sparsity level of 45.47%, the execution time reduces to 31.35 ms, resulting in a 1.86× speedup compared to the full mask. In Table 2, the effect of increasing threshold $r$ on speedup is evident. As $r$ increases, the sparsity grows, leading to a greater reduction in computation time and a corresponding increase in speedup. For example, with $r = 0.050$, the sparsity reaches 37.78%, achieving a speedup of 1.64×. When $r$ is further increased to 0.400, the sparsity level rises to 55.07%, and the speedup improves to 2.25×. This positive correlation between $r$, sparsity, and speedup highlights the efficiency gains that can be achieved by leveraging higher sparsity levels.

Table 1: Speedup with different masks.

| Frames | Mask | Sparsity (%) | Time(ms) | Speedup |
|---|---|---|---|---|
| 29 | full | 0.00 | 58.36 | 1.00× |
| | 4:4 | 17.60 | 46.52 | 1.25× |
| | 3:5 | 29.88 | 40.08 | 1.46× |
| | 2:6 | 45.47 | 31.35 | 1.86× |
| | 1:7 | 64.38 | 20.65 | 2.83× |
| 93 | full | 0.00 | 523.61 | 1.00× |
| | 12:12 | 21.51 | 397.72 | 1.32× |
| | 8:16 | 40.30 | 303.90 | 1.72× |
| | 6:18 | 51.88 | 244.13 | 2.14× |
| | 4:20 | 64.98 | 179.74 | 2.91× |
| | 3:21 | 72.05 | 142.77 | 3.67× |

Table 2: Speedup with different threshold $r$.

| Frames | $r$ | Sparsity (%) | Time(ms) | Speedup |
|---|---|---|---|---|
| 29 | full | 0.00 | 58.36 | 1.00× |
| | 0.025 | 23.51 | 43.50 | 1.34× |
| | 0.050 | 37.78 | 35.58 | 1.64× |
| | 0.100 | 45.08 | 31.54 | 1.85× |
| | 0.200 | 51.55 | 27.91 | 2.09× |
| | 0.400 | 55.07 | 25.96 | 2.25× |
| 93 | full | 0.00 | 523.61 | 1.00× |
| | 0.150 | 38.02 | 317.56 | 1.65× |

### 4.2.2 EFFICIENT-VDIT SPEEDUP IN DISTRIBUTED SETTING

EFFICIENT-VDIT utilize sparse attention and consistency distillation to achieve speedup. These methods are orthogonal to the recent advances in distributed systems, mainly sequence parallelism based solution in LLMs (Liu et al., 2023a; Li et al., 2024a; Jacobs et al., 2023) and model parallelism (or with hybrid sequence parallelism) based solution in diffusion Transformers (Li et al., 2024c; Wang et al., 2024a; Chen et al., 2024d). We consider sequence parallelism in this section for is simplicity and empirical lower overhead (Li et al., 2024a;c; Xue et al., 2024).

**Implementation** We utilize the All-to-All communication primitives to implement sequence parallelism (Jacobs et al., 2023). In the attention computation, the system partitions the operations along the head dimension while keeping the entire sequence intact on each GPU, allowing a simple implementation of EFFICIENT-VDIT by applying the same attention mask as in the one GPU setting [2]. As a result, EFFICIENT-VDIT is natively compatible with All-to-All sequence parallelism.

We conduct a scaling experiment with sequence parallelism on 4x A100-SXM 80GB GPUs, interconnected with NVLink. We observe a speedup of $3.68\times$ - $3.91\times$ for 29 and 93 frames generation on 4 GPUs, which is close to a theoretical speedup of $4\times$ (Table 3). If reported 29 frames generation on multi-GPUs, Ours$_{r=0.100}$ can achieve 25.8x speedup on 4 GPUs and 13.0x speedup on 2 GPUs.

Table 3: EFFICIENT-VDIT with sequence parallelism. Time as wall-clock-time per step.

| Frames | # GPUs | Time (s) | Speedup |
|---|---|---|---|
| 29 | 1 | 5.56 | $1.00\times$ |
| | 2 | 2.98 | $1.87\times$ |
| | 4 | 1.52 | $3.68\times$ |
| 93 | 1 | 39.06 | $1.00\times$ |
| | 2 | 20.00 | $1.95\times$ |
| | 4 | 10.02 | $3.91\times$ |

Table 4: Results on Open-Sora-Plan with 93 frames and 720p resolution. We select motion smoothness and temporal flickering from VBench as they measure frame transition, which are crucial for sparse attention methods.

| Model | Motion Smoothness | Temporal Flickering | Speedup |
|---|---|---|---|
| Base | 99.15% | 98.76% | $1.00\times$ |
| MLCD | **99.30%** | 99.22% | $5.00\times$ |
| Ours$_{r=0.150}$ | 99.08% | **99.31%** | **7.40**$\times$ |

### 4.3 VIDEO QUALITY BENCHMARK

Table 5: Open-Sora-Plan with 29 frames and 720p resolution results on VBench and CD-FVD. '$r$=0.1' indicates that this checkpoint is trained using the layerwise search strategy described in Algorithm 1, with a threshold of $r$=0.1. We selects some dimensions for analysis, with the remaining dimensions provide in the Table 8.

| Model | Final Score ↑ | Aesthetic Quality | Dynamic Degree | Motion Smoothness | Temporal Flickering | Object Class | Subject Consistency | Imaging Quality | CD-FVD ↓ | Speedup |
|---|---|---|---|---|---|---|---|---|---|---|
| Base | 76.12% | 58.34% | 34.72% | 99.43% | 99.28% | 64.72% | 98.45% | 64.75% | 172.64 | $1.00\times$ |
| MLCD | 76.81% | 58.92% | 41.67% | 99.41% | 99.42% | 63.37% | 98.37% | 65.55% | 190.50 | $5.00\times$ |
| Ours$_{r=0.025}$ | **76.14%** | 57.21% | 52.78% | **99.37%** | 99.49% | **60.36%** | **98.26%** | **58.90%** | 186.84 | $5.85\times$ |
| Ours$_{r=0.050}$ | 76.01% | **57.57%** | 58.33% | 99.15% | **99.56%** | 58.70% | 97.58% | 56.86% | 195.55 | $6.60\times$ |
| Ours$_{r=0.100}$ | 76.00% | 56.59% | 63.89% | 99.13% | 99.54% | 57.12% | 97.73% | 54.88% | 204.13 | $7.05\times$ |
| Ours$_{r=0.200}$ | 75.02% | 55.71% | 59.72% | 99.03% | 99.50% | 55.22% | 97.28% | 54.07% | 223.75 | $7.50\times$ |
| Ours$_{r=0.400}$ | 75.30% | 55.79% | **65.28%** | 98.93% | 99.46% | 54.98% | 97.71% | 54.36% | 231.68 | **7.80**$\times$ |

In this section, we first evaluate EFFICIENT-VDIT with layerwise searching on CD-FVD and VBench (Huang et al., 2024; Ge et al., 2024). We compare with the baseline of the original Open-Sora-Plan 1.2 model, and the model we obtain only using the MLCD method. We then conduct two ablation experiments to understand the effectiveness of the MLCD method, and our layerwise searching algorithm.

Table 5 demonstrates the main result of the 29 frames model. In VBench, We find that the results of all our search models are within 1% final score against the Base model with no noticeable drop in several key dimensions. At higher acceleration ratios, such as Ours$_{r=0.400}$, the model maintains

---

[2]The difference is that the attention mask is applied to fewer number of attention heads.

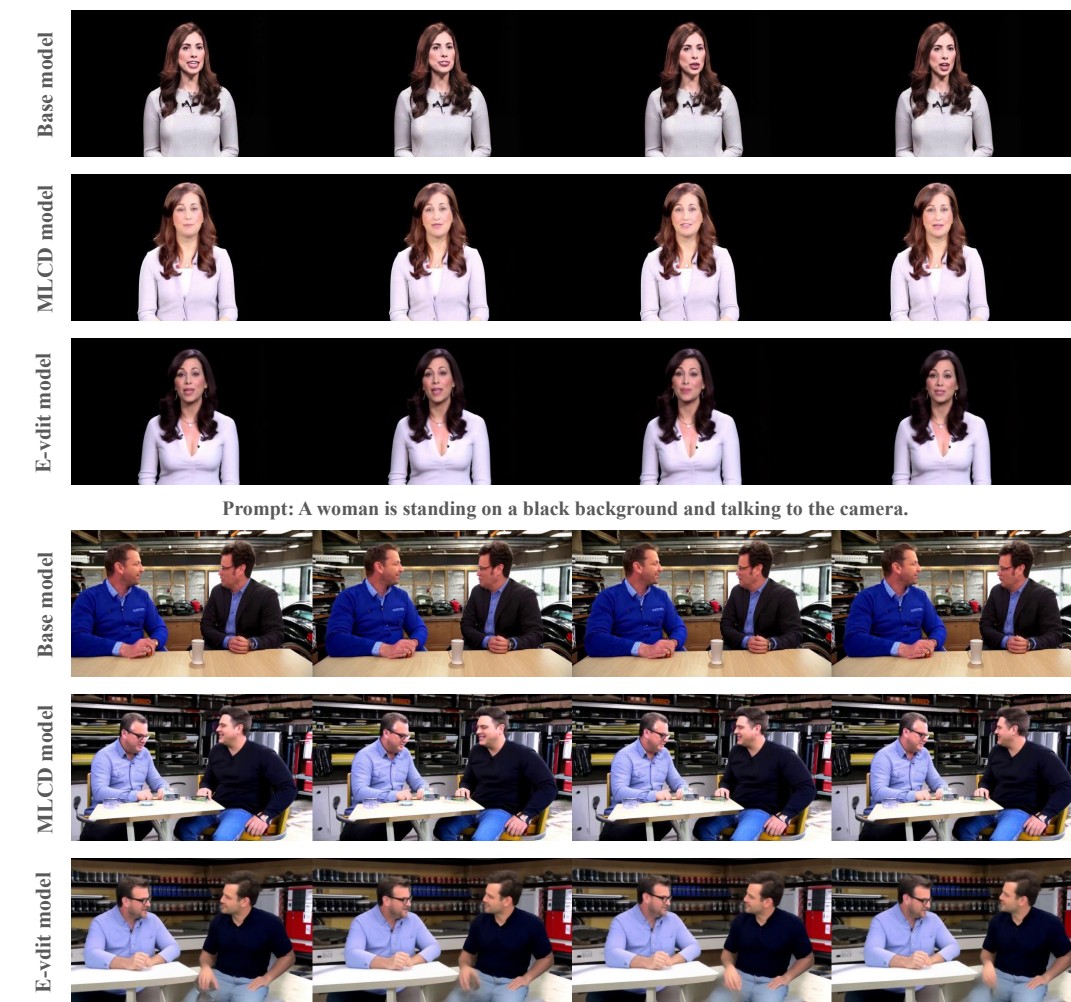

**Prompt: A woman is standing on a black background and talking to the camera.**

**Prompt: Two men are sitting at a table in a garage and talking to each other.**

Figure 5: Qualitative samples of our models. We compare the generation quality between the base model, MLCD model, and after knowledge distillation. Frames shown are equally spaced samples from the generated video. EFFICIENT-VDIT is shortened as 'E-vdit' for simplicity. More samples can be found in Appendix E.

stable performance, with minimal deviations from the Base model, demonstrating the robustness of our approach while achieving significant speedups. However, we note that the imaging quality and subject class are lower than those of the base model. The reason why the VBench score remains within 1% difference is that our model improves the dynamic degree. With more sparsity, our pipeline has the characteristics of being able to capture richer motions between frames, but trading off some degrees of imaging quality and subject class accuracy.

In CD-FVD, our models with smaller acceleration ratios achieve better scores than MLCD model. For example, $\text{Ours}_{r=0.025}$ achieves a score of 186.84 with a speedup of $5.85\times$, outperforming the MLCD model. As the acceleration ratio increases, the score degrades as expected. $\text{Ours}_{r=0.400}$ reaches a score of 231.68 with a speedup of $7.80\times$, showing a trade-off between acceleration and performance. Our models maintain performance with minimal performance drop and achieve a significant speedup. In table 4, we show the effectiveness of EFFICIENT-VDIT in a subset of VBench for 93 frames. We observe a similar conclusion that we achieve $7.4\times$ speedup.

**Effect of MLCD** We conduct tests on VBench and CD-FVD, first comparing the differences between the Base model and the MLCD model, and then evaluating the compatibility of CM with the attention mask. As shown in Table 6, the MLCD model performs as well as or better than the Base model across most dimensions on VBench, achieving an overall VBench score of 76.81%.

Due to the MLCD model requiring fewer sampling steps than the Base model, it achieves a $5.00\times$ speedup. Furthermore, we observe that the MLCD model, even after undergoing knowledge distillation, maintains performance without any drop in quality. The VBench score and CD-FVD trends are consistent, indicating that the MLCD model supports attention mask operations effectively, similar to the original model. Therefore, the MLCD model continues to deliver high-quality performance while offering significant acceleration benefits.

**Effect of Layerwise Search** We conduct tests on VBench and CD-FVD, selecting the MLCD model as the baseline. We compare applying a uniform mask across all layers (e.g., 4:4, 3:5) with the layerwise mask from Algorithm 1. As shown in Table 7, in VBench, using the layerwise mask with ($r = 0.025, 0.050, 0.100$) achieve a score exceeding 76.00%, significantly outperforming the results without layerwise masking, while also providing a better speedup ($7.05\times$ vs. $5.80\times$). In CD-FVD, the layerwise mask consistently results in scores below 250. However, as sparsity increases, the score without layerwise masking exceeds 250, indicating a decrease in video generation quality. Therefore, the layerwise approach enhances the quality of generated videos.

Table 6: Ablation experiments on the effect of MLCD.

| Model | Final Score ↑ | Aesthetic Quality | Dynamic Degree | Motion Smoothness | Temporal Flickering | Object Class | Subject Consistency | Imaging Quality | CD-FVD ↓ | Speedup |
|---|---|---|---|---|---|---|---|---|---|---|
| Base | 76.12% | 58.34% | 34.72% | **99.43%** | 99.28% | 64.72% | **98.45%** | 64.75% | 172.64 | 1.00× |
| Base$_{4:4}$ | 76.57% | 58.64% | 43.06% | 99.38% | 99.20% | **66.38%** | 98.26% | 63.56% | **171.62** | 1.16× |
| Base$_{3:5}$ | 75.53% | 55.47% | 58.33% | 99.01% | 98.96% | 62.26% | 97.42% | 59.67% | 197.35 | 1.26× |
| Base$_{2:6}$ | 76.33% | 57.14% | 56.94% | 99.06% | 99.02% | 56.17% | 97.58% | 61.10% | 201.61 | 1.45× |
| Base$_{1:7}$ | **77.15%** | 57.53% | **75.00%** | 98.67% | 98.66% | 60.68% | 96.96% | 61.91% | 322.28 | 1.77× |
| MLCD | 76.81% | **58.92%** | 41.67% | 99.41% | 99.42% | 63.37% | 98.37% | **65.55%** | 190.50 | 5.00× |
| MLCD$_{4:4}$ | 75.90% | 57.84% | 50.00% | 99.38% | **99.50%** | 63.03% | 98.21% | 58.47% | 175.47 | 5.80× |
| MLCD$_{3:5}$ | 75.41% | 57.19% | 43.06% | 99.36% | 99.50% | 57.04% | 98.12% | 58.84% | 190.92 | 6.30× |
| MLCD$_{2:6}$ | 75.23% | 57.45% | 44.44% | 99.29% | 99.48% | 54.59% | 98.37% | 57.35% | 213.72 | 7.25× |
| MLCD$_{1:7}$ | 75.84% | 56.83% | 63.89% | 98.99% | 99.23% | 52.77% | 97.54% | 56.42% | 294.09 | **8.85×** |

Table 7: Ablation experiments on the effect of our layerwise searching algorithm.

| Model | Final Score ↑ | Aesthetic Quality | Dynamic Degree | Motion Smoothness | Temporal Flickering | Object Class | Subject Consistency | Imaging Quality | CD-FVD ↓ | Speedup |
|---|---|---|---|---|---|---|---|---|---|---|
| MLCD | **76.81%** | **58.92%** | 41.67% | **99.41%** | 99.42% | **63.37%** | **98.37%** | **65.55%** | 190.50 | 5.00× |
| MLCD$_{4:4}$ | 75.90% | 57.84% | 50.00% | 99.38% | 99.50% | 63.03% | 98.21% | 58.47% | **175.47** | 5.80× |
| MLCD$_{3:5}$ | 75.41% | 57.19% | 43.06% | 99.36% | 99.50% | 57.04% | 98.12% | 58.84% | 190.91 | 6.30× |
| MLCD$_{2:6}$ | 75.23% | 57.45% | 44.44% | 99.29% | 99.48% | 54.59% | 98.37% | 57.35% | 213.71 | 7.25× |
| MLCD$_{1:7}$ | 75.84% | 56.83% | 63.89% | 98.99% | 99.23% | 52.77% | 97.54% | 56.42% | 294.09 | **8.85×** |
| Ours$_{r=0.025}$ | 76.14% | 57.21% | 52.78% | 99.37% | 99.49% | 60.36% | 98.26% | 58.90% | 186.84 | 5.85× |
| Ours$_{r=0.050}$ | 76.01% | 57.57% | 58.33% | 99.15% | **99.56%** | 58.70% | 97.58% | 56.86% | 195.55 | 6.60× |
| Ours$_{r=0.100}$ | 76.00% | 56.59% | 63.89% | 99.13% | 99.54% | 57.12% | 97.73% | 54.88% | 204.13 | 7.05× |
| Ours$_{r=0.200}$ | 75.02% | 55.71% | 59.72% | 99.03% | 99.50% | 55.22% | 97.28% | 54.07% | 223.75 | 7.50× |
| Ours$_{r=0.400}$ | 75.30% | 55.79% | **65.28%** | 98.93% | 99.46% | 54.98% | 97.71% | 54.36% | 231.68 | 7.80× |

## 4.4 QUALITATIVE RESULT

As illustrated in Figure 5, we compare the video results generated by three methods: the original model, after applying MLCD, and after knowledge distillation. The generation settings are consistent with those in Table 5, demonstrating that both the MLCD and knowledge distillation methods maintain the original quality and details. More qualitvative samples are listed in Appendix E.

## 5 CONCLUSION

In this paper, we first describe the phenomenon of *Attention Tile*, and dive into its characteristics of repetitive, large diagonals, locality, and data independent. Then we describe a class of sparse attention pattern tailored to address the efficiency problem in *Attention Tile*. Lastly, we introduce our overall framework that leveraged this class of sparse attention, which further leverages multi-step consistency distillation, layerwise searching, and knowledge distillation for faster generation and high performance. Experiments on two varaints of the Open-Sora-Plan model has demonstrated that our method can achieve similar performance, with 0.1% the pre-training data, and up to $7.8\times$ speedup. Further ablation study has shown that our method can be natively integrated with advanced parallelism method to achieve further speedup.

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

## A  EXTENED LAYERWISE SEARCH ALGORITHM

In this section, we explore how to balance the trade-off between inference speedup and output image quality. Intuitively, as the attention map becomes sparser, the inference time decreases, but the output image quality also degrades. With this model, we can answer the key question: **Given a target speedup or inference time, how can we achieve the highest possible image quality?**

This problem is well-suited to latency constrained case because, in real-world applications, speedup can be precisely measured. Adjusting the generation quality within these constraints is therefore meaningful. Additionally, solving this problem allows us to approximate continuous speedup ratios as closely as possible using discrete masks, further validating the robustness of our algorithm.

## A.1 Estimation and Quantitative Analysis

The inference time can be quantitatively computed. Given time limitation $T_{\text{target}}$. Suppose we have a series of masks $M_1, M_2, \ldots, M_k$. For each mask, we can pre-profile its runtime as $T_1, T_2, \ldots, T_k$. If layer $j$ uses mask $a_j \in [1, k]$, the total inference time is given by $T = \sum_j T_{a_j} \leq T_{\text{target}}$.

On the other hand, quantifying image quality is challenging. To address this, we make an assumption: the impact of different layers on image quality is additive. We use the loss as the value function, representing the output image quality as $\mathcal{L} = \sum_j \mathcal{L}_{j,a_j}$, where $\mathcal{L}_{j,a_j}$ denotes the loss value when layer $j$ uses mask type $a_j$.

## A.2 Lagrangian Relaxation Method

By introducing a Lagrange multiplier $\lambda$, we construct the Lagrangian function:

$$L(\lambda) = \sum_j \mathcal{L}_{j,a_j} + \lambda \left( \sum_j T_{a_j} - T_{\text{target}} \right). \tag{4}$$

Our goal is to minimize $L(\lambda)$, that is:

$$\min_{a_j} L(\lambda) = \min_{a_j} \left( \sum_j \mathcal{L}_{j,a_j} + \lambda \sum_j T_{a_j} \right) - \lambda T_{\text{target}}. \tag{5}$$

Since $T_{\text{target}}$ is a constant, the optimization problem can be simplified into independent subproblems for each layer $j$:

$$\min_{a_j} \left( \mathcal{L}_{j,a_j} + \lambda T_{a_j} \right). \tag{6}$$

## A.3 Lagrangian Subgradient Method

**Input:** Initial Lagrange multiplier $\lambda^{(0)}$, learning rate $\alpha_t$, maximum iterations $N$.
**Output:** Approximate optimal solution $\{a_j\}$ and Lagrange multiplier $\lambda$.

1. **Initialization:** Set iteration counter $t = 0$.

2. **While** $t < N$ and not converged:

   (a) **Step 1: Solve Subproblems**
   For each layer $j$, solve the subproblem:

   $$a_j^{(t)} = \arg\min_{a_j} \left( \mathcal{L}_{j,a_j} + \lambda^{(t)} T_{a_j} \right). \tag{7}$$

   (b) **Step 2: Calculate Subgradient**
   Compute the subgradient:

   $$g^{(t)} = \sum_j T_{a_j^{(t)}} - T_{\text{target}}. \tag{8}$$

   (c) **Step 3: Update Lagrange Multiplier**
   Update $\lambda$ using the subgradient:

   $$\lambda^{(t+1)} = \lambda^{(t)} + \alpha_t g^{(t)}. \tag{9}$$

   (d) Update $t = t + 1$.

**Output:** Return the approximate solution $\{a_j\}$ and the final Lagrange multiplier $\lambda$.

## B    FLEXATTENTION IMPLEMENTATION DETAILS

The attention we design can be efficiently implemented by the native block-wise computation design in FlexAttention. Compared to a dynamic implementations, our computations are static, allowing us to leverage static CUDA graphs for capturing or use PyTorch's `compile=True` feature.

FlexAttention employs a block-based mechanism that allows for efficient handling of sparse attention patterns. Specifically, when an empty block is encountered, the module automatically skips the attention computation, leveraging the sparsity in the attention matrix to accelerate calculations. The ability to skip computations in this manner results in significant speedups while maintaining efficient memory usage.

Additionally, FlexAttention is optimized by avoiding the need to materialize the entire mask. This mechanism enables FlexAttention to operate efficiently on large-scale models without incurring significant memory costs. For example, the additional memory usage of a model with 32 layers and a 29 frames mask is only 0.278GB, while a 93 frames mask requires 0.715GB of additional memory, which is considered minimal for large-scale models. By not needing to store or process the full mask, we save both memory and computation time, leading to improved performance, especially in scenarios where the attention matrix is highly sparse.

## C    SUPPLEMENTAL VBENCH EVALUATION

Table 8: Supplemental VBench evaluation for main result.

| Model | Multiple Objects | Human Action | Color | Spatial Relationship | Scene | Appearance Style | Temporal Style | Overall Consistency | Background Consistency |
|---|---|---|---|---|---|---|---|---|---|
| Base | 23.25% | 54.00% | 94.47% | 43.49% | 18.60% | 19.88% | 18.45% | 19.69% | 97.64% |
| MLCD | 19.21% | 56.00% | 94.12% | 40.57% | 22.67% | 20.46% | 18.21% | 19.77% | 97.98% |
| Ours$_{r=0.025}$ | 18.83% | 55.00% | **96.25%** | **46.02%** | 12.35% | **20.31%** | 18.17% | 19.11% | 97.70% |
| Ours$_{r=0.050}$ | 11.74% | **58.00%** | 92.11% | 39.81% | **22.31%** | 20.25% | 17.71% | **19.45%** | **97.71%** |
| Ours$_{r=0.100}$ | **18.98%** | 56.00% | 93.65% | 43.88% | 15.77% | 20.20% | 17.98% | 19.29% | 97.55% |
| Ours$_{r=0.200}$ | 17.99% | 53.00% | 51.82% | 36.14% | 13.88% | 20.29% | 17.97% | 18.97% | 97.62% |
| Ours$_{r=0.400}$ | 15.32% | 54.00% | 92.64% | 37.05% | 12.06% | 20.24% | **18.19%** | 19.22% | 97.66% |

Table 9: Supplemental VBench evaluation result for base model and MLCD ablation experiment.

| Model | Multiple Objects | Human Action | Color | Spatial Relationship | Scene | Appearance Style | Temporal Style | Overall Consistency | Background Consistency |
|---|---|---|---|---|---|---|---|---|---|
| Base | 23.25% | 54.00% | **94.47%** | 43.49% | 18.60% | 19.88% | **18.45%** | 19.69% | 97.64% |
| Base$_{4:4}$ | **32.01%** | 55.00% | 90.94% | **45.42%** | 17.30% | 20.21% | 18.41% | 19.48% | 97.17% |
| Base$_{3:5}$ | 15.85% | 53.00% | 88.88% | 44.38% | 14.53% | 20.13% | 17.46% | 18.43% | 97.28% |
| Base$_{2:6}$ | 21.65% | 56.00% | 93.27% | 49.90% | 18.31% | 19.87% | 18.23% | 18.94% | 97.27% |
| Base$_{1:7}$ | 17.76% | 54.00% | 93.02% | 44.75% | 19.99% | 19.95% | 18.25% | 19.41% | 97.30% |
| MLCD | 19.21% | **56.00%** | 94.12% | 40.57% | **22.67%** | **20.46%** | 18.21% | **19.77%** | **97.98%** |
| MLCD$_{4:4}$ | 22.79% | 53.00% | 92.69% | 39.80% | 17.51% | 19.89% | 18.32% | 19.06% | 97.30% |
| MLCD$_{3:5}$ | 22.10% | 50.00% | 90.82% | 43.48% | 21.44% | 19.97% | 17.68% | 19.75% | 97.47% |
| MLCD$_{2:6}$ | 18.60% | 53.00% | 92.52% | 43.36% | 16.21% | 19.89% | 17.84% | 20.12% | 97.70% |
| MLCD$_{1:7}$ | 16.92% | 53.00% | 91.92% | 43.27% | 17.22% | 19.94% | 18.56% | 19.85% | 97.45% |

## D    ABLATION STUDY OF KNOWLEDGE DISTILLATION

### D.1    ABLATION STUDY OF KNOWLEDGE DISTILLATION AND CONSISTENCY DISTILLATION ORDER

We claim that knowledge distillation and consistency distillation are orthogonal processes. To verify this, we conducted an ablation experiment on the distillation order. We first applied attention distillation based on the original model, then used this model to perform multi-step latent consistency distillation (MLCD). The results in Table 11 support our hypothesis, showing minimal differences in VBench and CD-FVD scores regardless of the distillation sequence. We also show qualitative samples in Figure 6 to illustrate the video quality.

Table 10: Supplemental VBench evaluation result for MLCD and layerwise knowledge distillation ablation experiment.

| Model | Multiple Objects | Human Action | Color | Spatial Relationship | Scene | Appearance Style | Temporal Style | Overall Consistency | Background Consistency |
|---|---|---|---|---|---|---|---|---|---|
| MLCD | 19.21% | 56.00% | 94.12% | 40.57% | **22.67%** | **20.46%** | 18.21% | 19.77% | **97.98%** |
| MLCD$_{4:4}$ | **22.79%** | 53.00% | 92.69% | 39.80% | 17.51% | 19.89% | 18.32% | 19.06% | 97.30% |
| MLCD$_{3:5}$ | 22.10% | 50.00% | 90.82% | 43.48% | 21.44% | 19.97% | 17.68% | 19.75% | 97.47% |
| MLCD$_{2:6}$ | 18.60% | 53.00% | 92.52% | 43.36% | 16.21% | 19.89% | 17.84% | **20.12%** | 97.70% |
| MLCD$_{1:7}$ | 16.92% | 53.00% | 91.92% | 43.27% | 17.22% | 19.94% | **18.56%** | 19.85% | 97.45% |
| Ours$_{r=0.025}$ | 18.83% | 55.00% | **96.25%** | **46.02%** | 12.35% | 20.31% | 18.17% | 19.11% | 97.70% |
| Ours$_{r=0.050}$ | 11.74% | **58.00%** | 92.11% | 39.81% | 22.31% | 20.25% | 17.71% | 19.45% | 97.71% |
| Ours$_{r=0.100}$ | 18.98% | 56.00% | 93.65% | 43.88% | 15.77% | 20.20% | 17.98% | 19.29% | 97.55% |
| Ours$_{r=0.200}$ | 17.99% | 53.00% | 51.82% | 36.14% | 13.88% | 20.29% | 17.97% | 18.97% | 97.62% |
| Ours$_{r=0.400}$ | 15.32% | 54.00% | 92.64% | 37.05% | 12.06% | 20.24% | 18.19% | 19.22% | 97.66% |

Table 11: VBench evaluation result for ablation study on distillation order for MLCD and layerwise knowledge distillation.

| Model | Final Score ↑ | Aesthetic Quality | Dynamic Degree | Motion Smoothness | Temporal Flickering | Object Class | Subject Consistency | Imaging Quality | CD-FVD ↓ |
|---|---|---|---|---|---|---|---|---|---|
| MLCD + KD | 76.00% | 56.59% | 63.88% | 99.13% | 99.54% | 57.12% | 97.73% | 54.88% | 204.13 |
| KD + MLCD | 75.50% | 56.38% | 54.16% | 99.12% | 99.40% | 54.67% | 97.71% | 57.97% | 203.52 |

| Model | Multiple Objects | Human Action | Color | Spatial Relationship | Scene | Appearance Style | Temporal Style | Overall Consistency | Background Consistency |
|---|---|---|---|---|---|---|---|---|---|
| MLCD + KD | 18.97% | 0.56% | 93.65% | 43.87% | 15.77% | 20.20% | 17.98% | 19.29% | 97.55% |
| KD + MLCD | 17.22% | 0.53% | 93.14% | 39.87% | 17.65% | 20.11% | 18.01% | 19.17% | 97.69% |

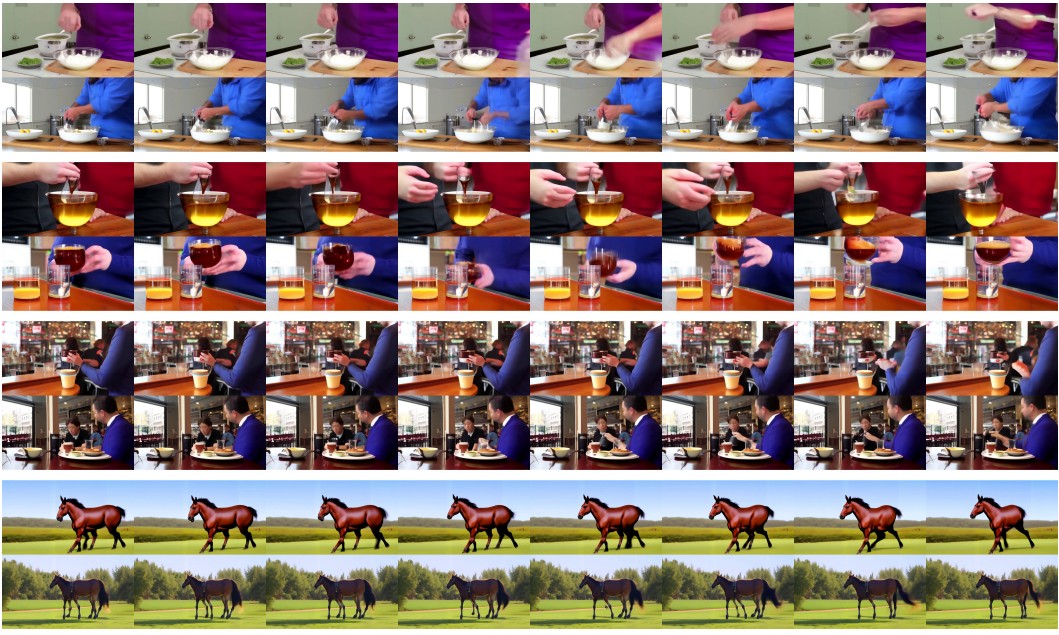

Figure 6: Qualitative samples of ablation of distillation order. sampled from VBench prompts. We show that both MLCD and EFFICIENT-VDIT model can simliar quality on these samples. In two consecutive videos, the top shows results from MLCD + CD model followed by KD + MLCD model.

## D.2 ABLATION STUDY OF ATTENTION DISTILL ON COGVIDEOX MODEL

We show that attention distillation also works well on the CogVideoX Yang et al. (2024b) model. CogVideoX is based on the MM-DiT architecture, where its attention module concatenates text tokens with video tokens, which differs from Open-Sora-Plan's cross attention module. This demonstrates that our method works effectively on both MM-DiT and cross attention architectures. Our experiments are conducted on the CogVideoX-5B model with 49-frame generation capability.

**Implementation Details** CogVideoX-5B is profiled using Algorithm 1. For training, the model is trained for a total of 10,000 steps, equivalent to 10 epochs of the dataset. The learning rate is set to 1e-7, and the gradient accumulation step is set to 1. The diffusion scale factor $\lambda$ is set to 1.

**Kernel Performance** We analyze the computation time for a single sparse attention kernel in Table 12. The results show that as sparsity increases, computation time decreases significantly. For instance, with a 2:11 attention mask, the execution time reduces to 15.16ms, achieving a 1.72× speedup compared to the full mask.

Table 12: CogvideoX-5B model speedup with different masks.

| Mask | Sparsity (%) | Time(ms) | Speedup |
|------|--------------|----------|---------|
| full | 0.00 | 26.03 | 1.00× |
| 1 | 14.50 | 24.12 | 1.08× |
| 2 | 29.29 | 23.68 | 1.10× |
| 3 | 38.30 | 20.51 | 1.27× |
| 4 | 48.66 | 17.77 | 1.47× |
| 6 | 60.15 | 14.08 | 1.85× |
| 12 | 74.11 | 9.99 | 2.60× |

**Evaluation** For quantitative analysis, we show the VBench evaluation results of the knowledge distillation model in Table 13. The results of our model are within 1% of the final score with no noticeable drop in several key dimensions. Our model achieves comparable performance to the original model. For qualitative analysis, we present sample visualizations in Figure 7 to demonstrate the video generation quality. These evaluations show that our method maintains similar video quality while achieving significant speedup, validating its effectiveness across different video diffusion model architectures.

Table 13: CogVideoX-5B with 49 frames and 480p resolution results on VBench. '$r$=4.0' indicates that this checkpoint was trained using the layerwise search strategy described in Algorithm 1, with a threshold of $r$=4.0.

| Model | Final Score ↑ | Aesthetic Quality | Dynamic Degree | Motion Smoothness | Temporal Flickering | Object Class | Subject Consistency | Imaging Quality | Speedup |
|-------|---------------|-------------------|----------------|-------------------|---------------------|--------------|---------------------|-----------------|---------|
| Base | 77.91% | 57.91% | 76.39% | 97.83% | 97.34% | 71.99% | 92.27% | 57.78% | 1.00× |
| Ours$_{r=5}$ | 77.15% | 51.18% | 86.11% | 96.67% | 97.18% | 77.06% | 90.89% | 55.75% | 1.34× |

| Model | Multiple Objects | Human Action | Color | Spatial Relationship | Scene | Appearance Style | Temporal Style | Overall Consistency | Background Consistency |
|-------|------------------|--------------|-------|----------------------|-------|------------------|----------------|---------------------|------------------------|
| Base | 48.62% | 84.00% | 86.71% | 48.47% | 38.01% | 22.99% | 23.22% | 26.13% | 95.01% |
| Ours$_{r=5}$ | 39.17% | 90.00% | 83.58% | 46.00% | 36.92% | 23.20% | 23.40% | 26.02% | 93.95% |

## E QUALITATIVE SAMPLES OF DYNAMIC SCENES AND LARGE-SCALE MOTION

We compare the generation quality between the base model, MLCD model, and after knowledge distillation. Our method (EFFICIENT-VDIT) is shortened as 'E-vdit' for simplicity. In Figure 8, we demonstrate that our model is capable of generating large-scale motion effects such as centralized radiating explosions. In Figure 9, we showcase our model's ability to generate dramatic physical movements, such as superhumans unleashing power in mid-air. In Figures 10, 11, 12, we show a series of samples from VBench prompts, demonstrating our model's motion generation capabilities and providing better insights into the VBench scoring results.

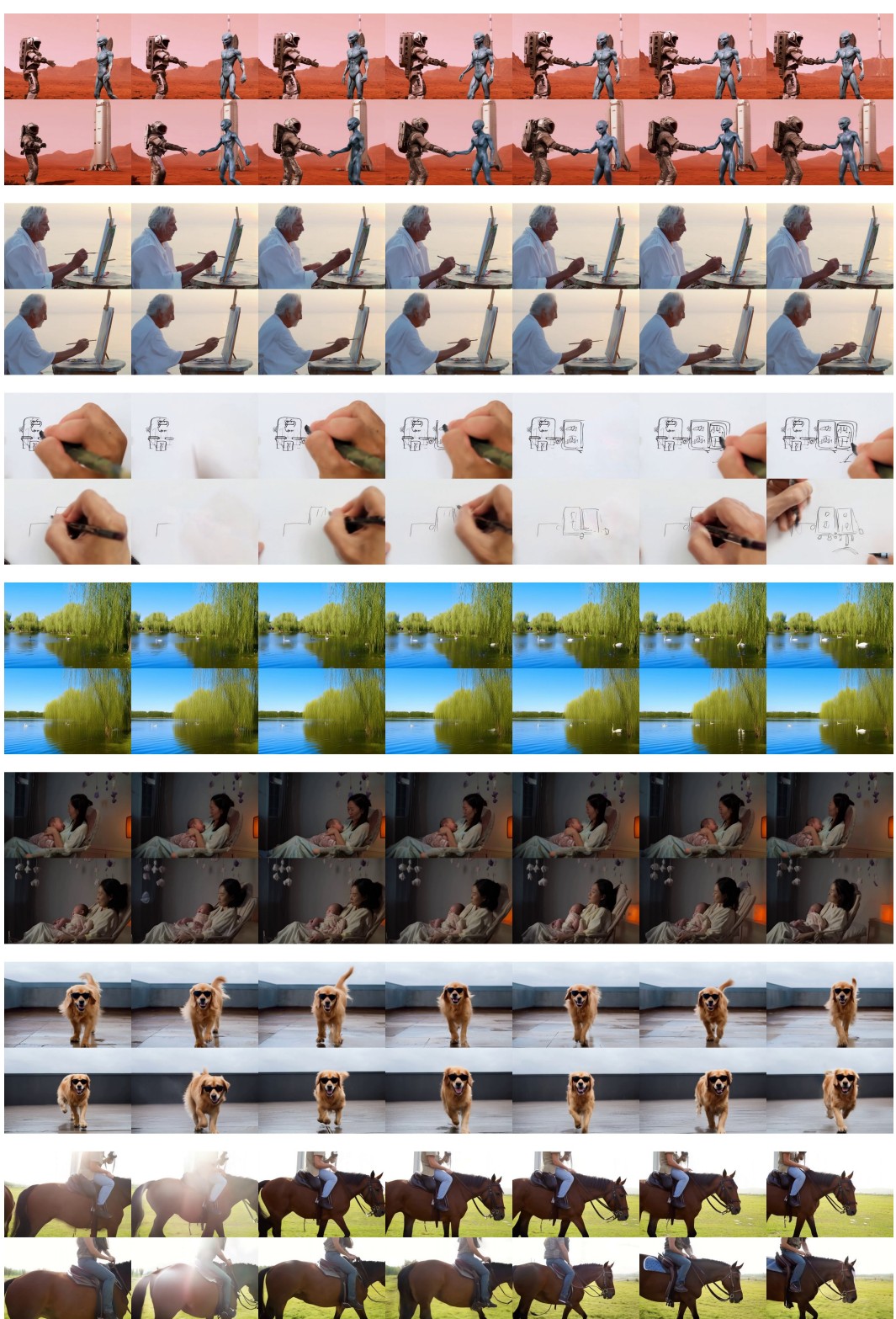

Figure 7: Qualitative samples of CogvideoX-5B Yang et al. (2024b) distillation from its sample prompts. We show that our attention distill is capable of MM-DiT model architecture. In two consecutive videos, the top shows results from the base model, followed by the distillation model.

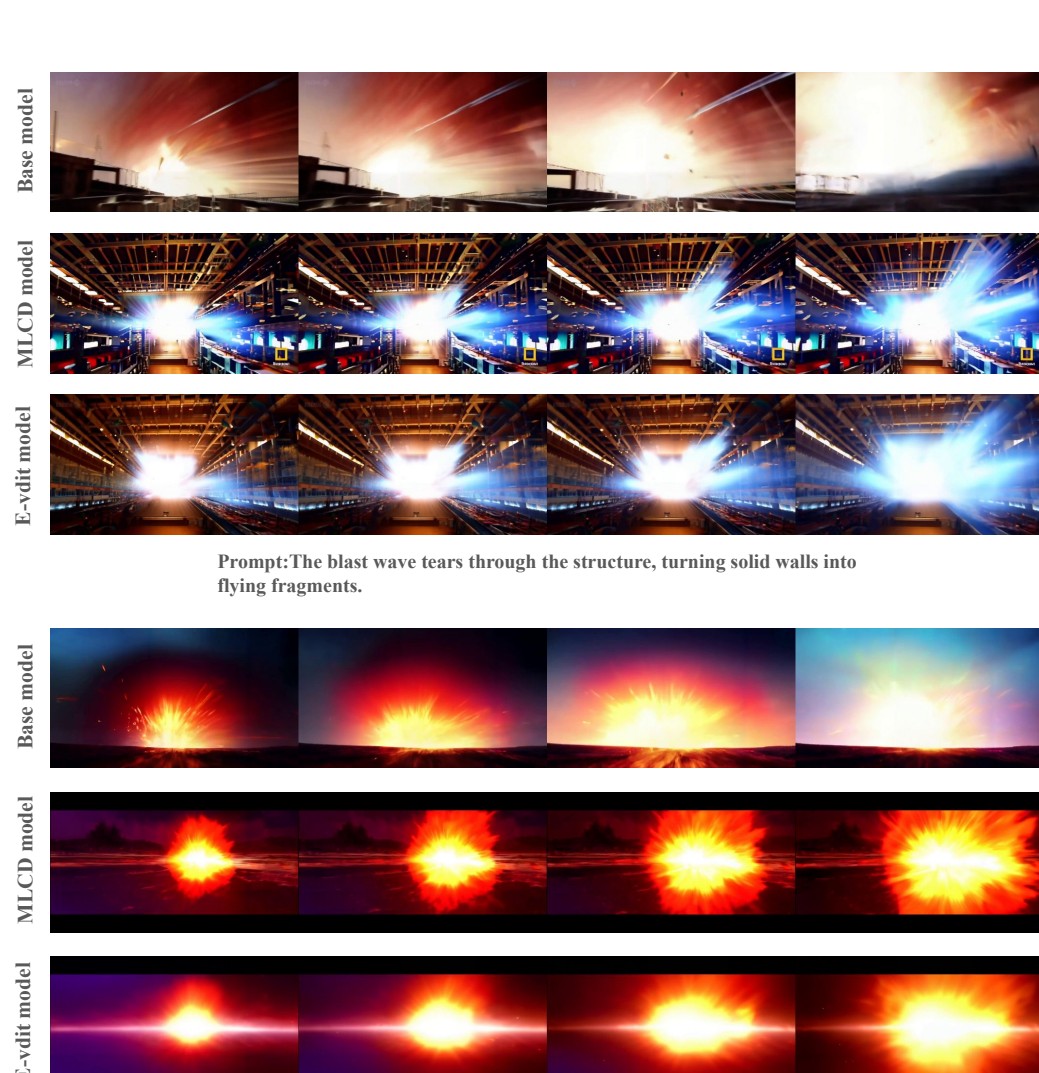

Prompt:The blast wave tears through the structure, turning solid walls into flying fragments.

Prompt:The firestorm implodes, creating a vacuum that pulls everything toward a single point.

Figure 8: Based on Open-Sora's examples Zheng et al. (2024) , we selected dynamic prompts featuring centralized explosions and radiating energy, demonstrating dramatic transitions from focal points to expansive environmental transformations, emphasizing large-scale motion.

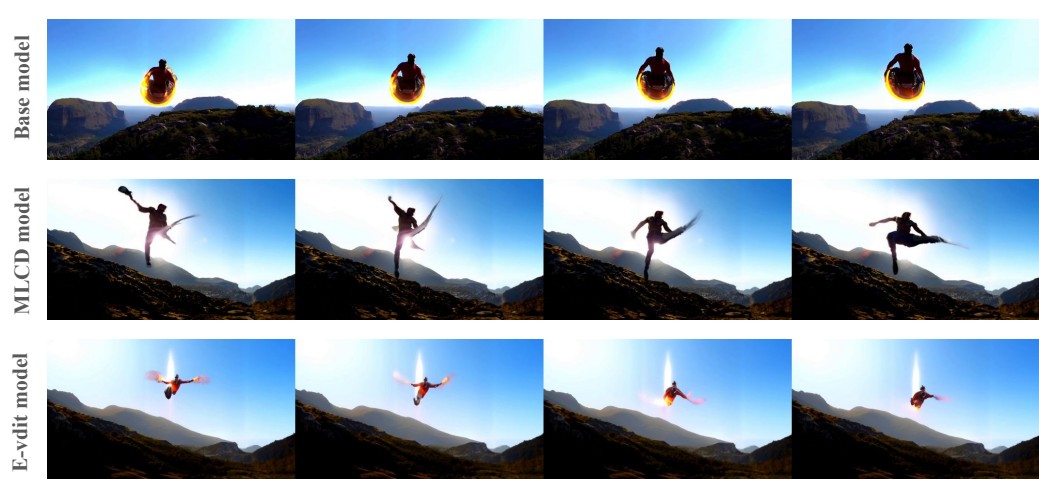

**Prompt:Superhuman unleashing ultimate power, body levitating, surrounding rocks suspended, energy ripples expanding, slow motion capture.**

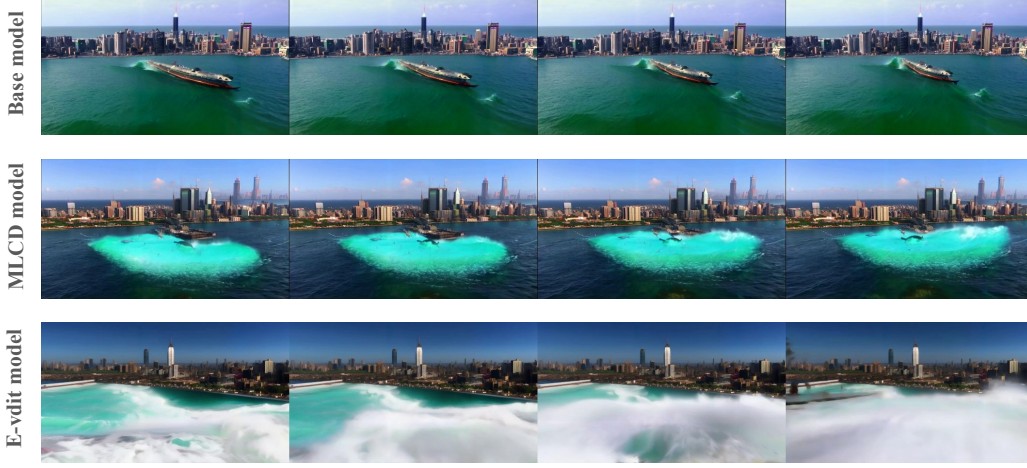

**Prompt:A kaiju emergence in a megacity bay. the camera sweeps around as a massive creature rises from the ocean, creating tsunami waves that crash through skyscrapers. defense mechanisms activate as the beast demonstrates impossible physics-defying abilities. buildings transform into combat modes. evacuation ships launch everywhere. the video focuses on the sheer scale of the creature and its reality-warping presence.**

Figure 9: Dynamic prompts featuring forceful physical movements (like kicking) and swirling environmental effects (like waves), transitioning from calm to intense states.

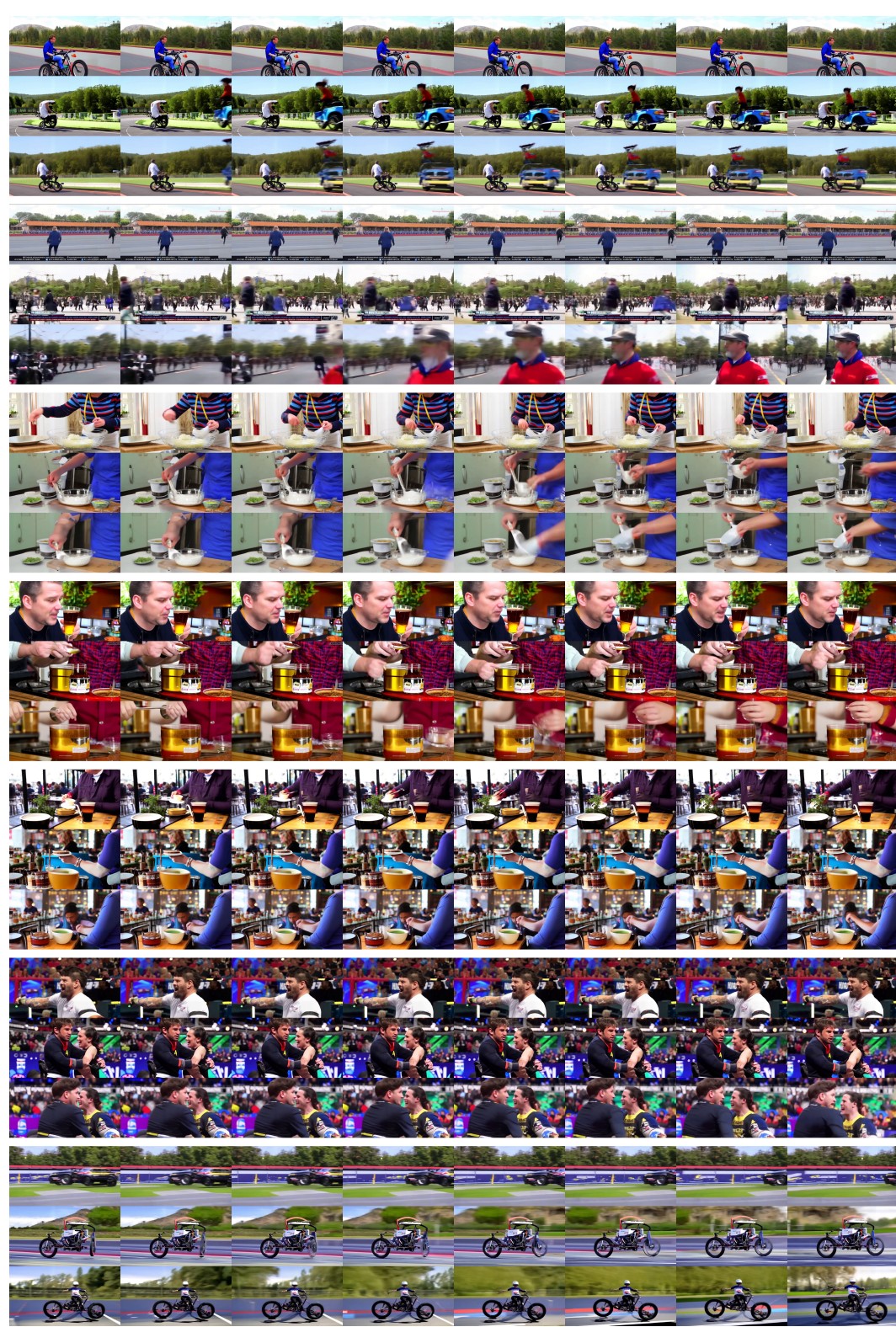

Figure 10: Qualitative samples of dynamic scenes from VBench prompts. We show that both MLCD and EFFICIENT-VDIT model can generate dynamic videos while maintaining video quality. In three consecutive videos, the top shows results from the base model, followed by the MLCD model, and the EFFICIENT-VDIT model.

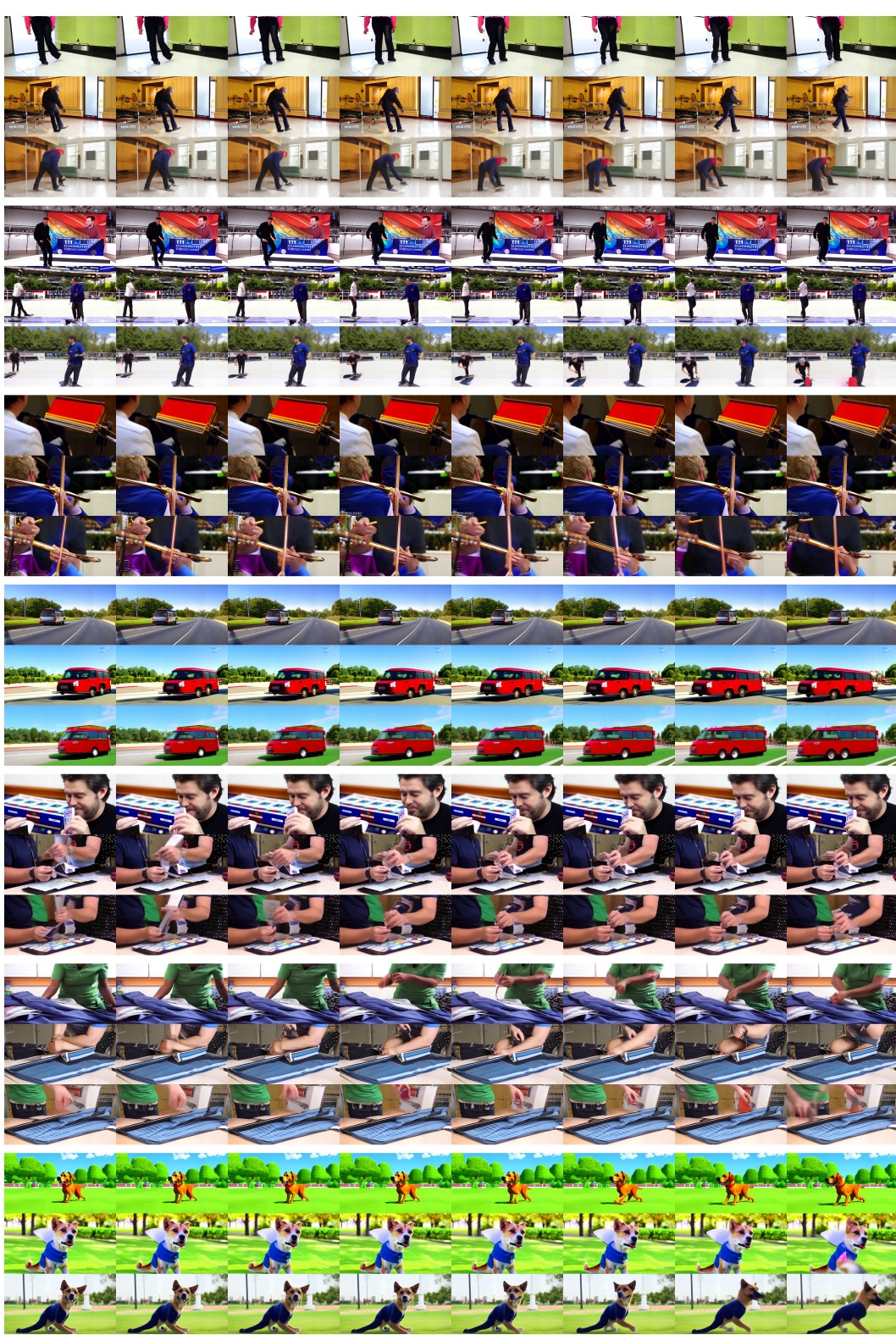

Figure 11: Qualitative samples of dynamic scenes from VBench prompts. We show that both MLCD and EFFICIENT-VDIT model can generate dynamic videos while maintaining video quality. In three consecutive videos, the top shows results from the base model, followed by the MLCD model, and the EFFICIENT-VDIT model.

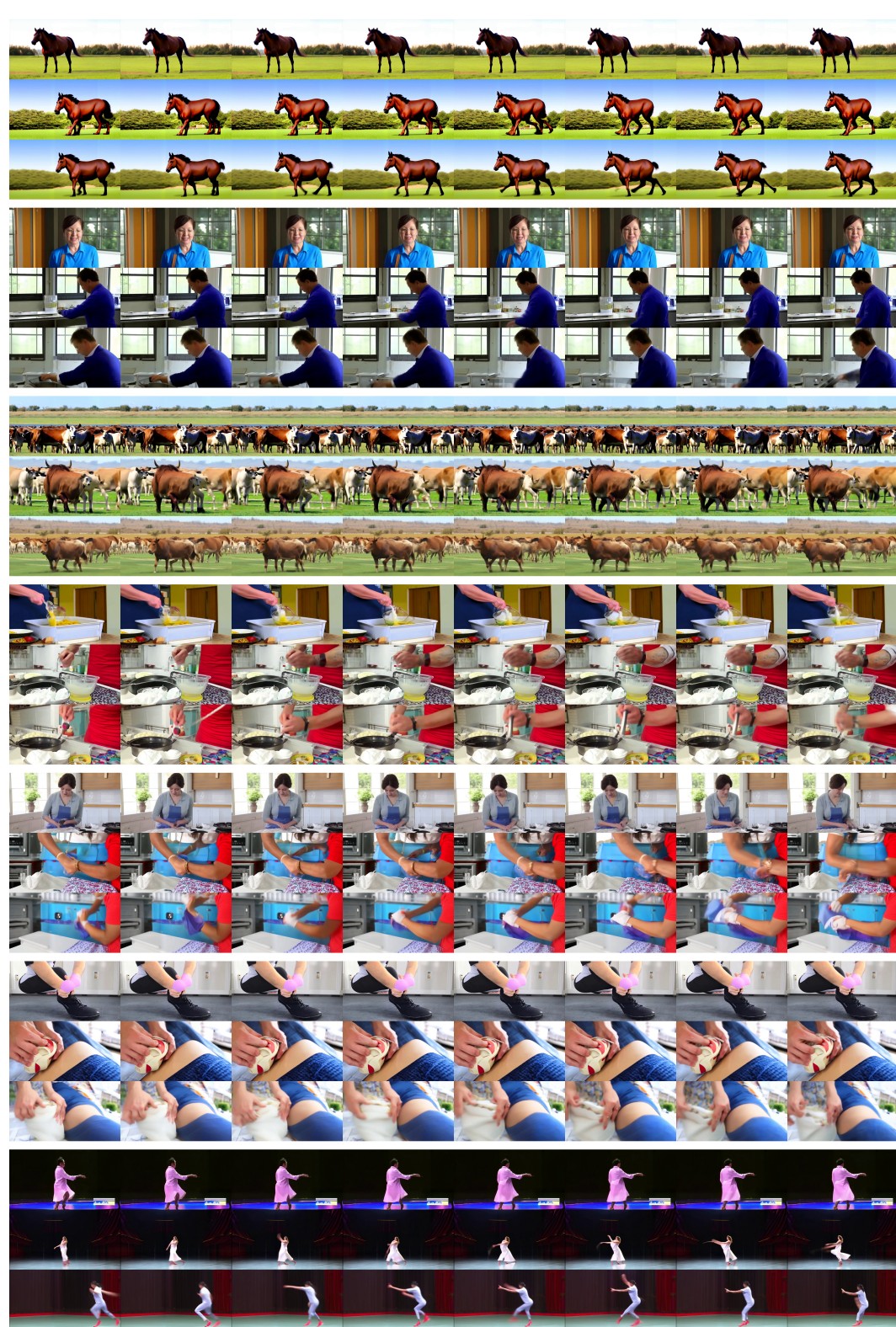

Figure 12: Qualitative samples of dynamic scenes from VBench prompts. We show that both MLCD and EFFICIENT-VDIT model can generate dynamic videos while maintaining video quality. In three consecutive videos, the top shows results from the base model, followed by the MLCD model, and the EFFICIENT-VDIT model.

