# OpenReview forum: "Efficient-vDiT: Efficient Video Diffusion Transformers With Attention Tile"
_ICLR.cc/2025/Conference — Submitted to ICLR 2025_

### Official Review · Reviewer_tExx · 2024-10-29

**Soundness:** 3
**Presentation:** 3
**Contribution:** 2
**Rating:** 6
**Confidence:** 4

**Summary:**

The authors propose a framework to speed up video generation using Video Diffusion Transformers by optimizing attention computation and reducing sampling steps. A repetitive attention tile pattern in 3D attention maps is identified which allows for sparse attention that lowers complexity. The framework uses a three-stage training pipeline: multi-step consistency distillation to reduce sampling steps, a layer-wise search for optimal sparse attention masks, and knowledge distillation to retain performance. This approach claims to achieve up to a 7.8× speedup in video generation with minimal quality loss.

**Strengths:**

The paper is well-written.

The computational complexity of video diffusion models presents a significant challenge, and the authors effectively highlight this issue and provide a good motivation for addressing it.

To tackle this, the solution provided by the authors of using a sparse attention map is interesting. Although thinking in this direction is not new, the way the authors motivate the solution and compute the attention maps is scientifically sound and has some novelty.

The computational speed-up achieved by the method looks impressive.

**Weaknesses:**

In the video generation literature, there are models that generate frames sequentially or follow an auto-regressive approach [1,2]. These models may be less computationally expensive than those using full 3D attention heads, yet there is no empirical or theoretical comparison with such models in the paper.

There should be an ablation study with the separate effects of sparse attention (without the MLCD) to understand each component in more detail.

The sampling distillation stage (Stage 1) is not really new, either technically or conceptually. There has been a line of work that provides a similar methodology [3,4], etc. It is not clear how different the proposed distillation is from the existing literature. The same can be said for the knowledge distillation in the final stage (Stage 3).

The paper has only two qualitative video generation results (or at least what I have found), of which only four frames are shown. There should be a lot more generated videos shown side by side to compare the method qualitatively.

[1] Diffusion forcing: Next-token prediction meets full-sequence diffusion. Chen et al. 2024.

[2] Diffusion models are real-time game engines. Valevski et al. 2024.

[3] MLCM: Multistep Consistency Distillation of Latent Diffusion Model. Xie et al. 2024.

[4] SCott: Accelerating Diffusion Models with Stochastic Consistency Distillation. Liu et al. 2024.

**Questions:**

What happens if the stages are switched, i.e., first obtain T_{sparse}, then T_{MCM}​ from T_{sparse}, and finally apply the knowledge distillation step?

Table 4 needs additional quantitative metrics like aesthetic quality, subject consistency, imaging quality, and FVD to provide a complete understanding of the effect of parallelization.

When comparing speed-up performance for parallelization, are the baseline models also trained with parallelization (Table 4)?

How does the proposed model achieve a lower FVD (Table 5) than the main base model, given that the proposed model is ultimately a distilled version of the main model?

How is the claim (lines 424 to 430) that model performance is within 1% of the base model accurate? It is evident that the numbers for imaging quality and subject class are significantly lower than those of the base model.

Ablation studies in Table 6 show that only MLCD can speed up the process by 5 to 8 times compared to the base model without significantly compromising quality. What is the justification, then, for the need for sparse attention maps on top of that?

It seems the main contribution is the sparse attention part. However, some doubts remain. Therefore, I can increase my rating if my questions and concerns in the weakness section and questions section are answered satisfactorily.

---

> ### Author Response · Authors · 2024-11-21
> **Response [Part 1/2]**
>
> We thank the reviewer for acknowledging the novelty of our attention map findings and the speedup of our approach. We summarize the questions here and respond to them individually:
>
> ## Q1: No comparison with sequential/autoregressive video generation models that might be computationally cheaper.
> A1: We thank the reviewer for discussing the literature on sequential/autoregressive video generation models. In addition to the mentioned literature, we also include more references in this line [1][2][3] and have updated the paper accordingly. A major line of these works simply invokes diffusion models in an auto-regressive loop. Our method accelerates a single diffusion forward pass, which is compatible with these methods. For instance, [1] and [2] can simply replace their original diffusion model with one that is tuned with our Efficient-vDiT method, and benefit from the speedup without needing to change the overall generation procedure.
>
> [1] Henschel, R., et al. (2024). StreamingT2V: Consistent, Dynamic, and Extendable Long Video Generation from Text. arXiv preprint arXiv:2403.14773.
>
> [2] Xiang, J., et al. (2024). Pandora: Towards General World Model with Natural Language Actions and Video States. arXiv preprint arXiv:2406.09455.
>
> [3] Zheng, Z., et al. (2024). Open-Sora: Democratizing Efficient Video Production for All. https://github.com/hpcaitech/Open-Sora
>
> ## Q2: Lacks detailed ablation study showing separate effects of sparse attention without MLCD and experiment of first T_sparse then T_MLCD.
> A2:
> 1. We updated the experiments with T_sparse first and then T_MLCD in Appendix D. We find that the T_sparse stage is orthogonal to the T_MLCD stage, observing only negligible differences in the VBench, and CD-FVD scores.
>
> | Distill Order | Aesthetic Quality | Dynamic Degree | Motion Smoothness | Temporal Flickering | Object Class | Subject Consistency | Imaging Quality | CD-FVD$\downarrow$ |
> |:---:|:---:|:---:|:---:|:---:|:---:|:---:|:---:|:---:|
> | MLCD+KD | 56.59% | 76.00% | 99.13% | 99.54% | 57.12% | 97.73% | 54.88% | 204.13 |
> | KD+MLCD | 56.38% | 75.50% | 99.13% | 99.40% | 54.67% | 97.71% | 57.97% | 203.52 |
>
> 2. In Tables 6 and 7 (upper section), we present the evaluation results for distillation works applied independently to both the base model and MLCD, demonstrating the isolated effect of sparse attention. In Table 6, the CD-FVD scores for Base_{3:5} and Base_{4:4} consistently remain below 200. However, as the attention sparsity increases, the CD-FVD score rises to 322.28, indicating a degradation in video generation quality. A similar trend is observed in Table 7, confirming that sparse attention has comparable effects when applied to both the base model and MLCD model.
>
>
> ## Q3: Both the sampling distillation (Stage 1) and knowledge distillation (Stage 3) lack technical novelty, as similar methodologies have been proposed in previous works [3,4]. The distinction between our proposed distillation and existing literature is unclear.
> A3: Our main novelty is in the discovery of the analysis of attention tile redundancy patterns, where we adapted existing distillation processes to leverage this observation for inference acceleration.
>
> ## Q4: Only shows two video examples with four frames each.
> A4: We have included updated video examples from VBench and examples similar to those on OpenSora's website featuring dynamic scenes in Appendix E, demonstrating our capability to handle rapid, large-scale motions.

---

> > ### Author Response · Authors · 2024-11-21
> > **Response [Part 2/2]**
> >
> > ## Q5: When comparing speed-up performance for parallelization, are the baseline models also trained with parallelization (Table 4)?
> > A5: Yes, the model is trained with sequence parallelism using 4 GPUs. We thank the reviewer for the feedback and will update these details in the paper. However, regarding inference, we would like to clarify that Table 4 compares inference speed on a single GPU. Whether models are run with sequence parallelism (Table 3) is orthogonal to model performance (Table 4). If reported 29 frames generation on multi-GPUs, $\text{Ours}_{r\text{=0.100}}$  can achieve 25.8x speedup on 4 GPUs and 13.0x speedup on 2 GPUs (Sec4.2.2).
> >
> > ## Q6: How does the proposed model achieve a lower FVD than the main base model?
> > A6: FVD values indeed have considerable variances. Based on the other reviewer’s advice, we update the FVD result using the Content-Debiased FVD (CD-FVD) result, which more effectively captures the temporal coherence of video quality, and the proposed model achieves a slighter higher CD-FVD score.
> >
> > | Model | FVD ↓ | Content-Debiased FVD ↓ |
> > |:---:|:---:|:---:|
> > | Base| 381.1 | 172.6 |
> > | MLCD | 438.1 | 190.5 |
> > | $\text{Ours}_{r\text{=0.025}}$ | 351.6 | 186.8 |
> > | $\text{Ours}_{r\text{=0.050}}$ | 357.4 | 195.6 |
> > | $\text{Ours}_{r\text{=0.100}}$ | 345.6 | 204.1 |
> > | $\text{Ours}_{r\text{=0.200}}$ | 356.9 | 223.8 |
> > | $\text{Ours}_{r\text{=0.400}}$ | 380.2 | 231.7 |
> >
> >
> >
> > [4] Songwei Ge et al., On the Content Bias in Fréchet Video Distance, in CVPR 2024.
> >
> >
> > ## Q7: 1% difference is not reasonable.The imaging quality and subject class are significantly lower than those of the base model.
> > A7:
> > We agree with the reviewer that the scores for imaging quality and subject class are lower than those of the base model. The reason why the VBench score remains within 1% difference is that our model improves the dynamic degree. With more sparsity, we noted that our pipeline has the characteristics of being able to capture richer motions between frames, but trading off some degrees of imaging quality and subject class accuracy. Quantitatively, we measure our models and variants across the overall VBench dimensions to justify the quality of our method. Qualitatively, we provide more samples from the VBench dataset in Appendix E to demonstrate that our method improves motion dynamics while maintaining acceptable imaging quality and subject class accuracy.
> >
> > ## Q8: Justification: Questions the necessity of sparse attention given MLCD's strong performance alone
> > A8: Consistency distillation is indeed a strong method to reduce the redundancy between diffusion steps. However, we’d like to point out that the attention tile phenomenon and our attention distillation pipeline are complementary to consistency distillation, as they address a new kind of redundancy we discovered in video diffusion models. In other words, redundancy arises from both repetitive diffusion sampling and 3D attention; MLCD mitigates the former, while attention distillation focuses on the latter. As demonstrated in Tables 1 and 2, attention distillation alone accelerates inference significantly as well. Our approach synergistically integrates these techniques to efficiently eliminate redundancy in 3D-DiT.

---

### Official Review · Reviewer_edPF · 2024-10-30

**Soundness:** 2
**Presentation:** 3
**Contribution:** 2
**Rating:** 5
**Confidence:** 3

**Summary:**

This paper proposed an efficient method for DiT based text-to-video generation. They found a unique pattern in the attention map of DiT based video generation diffusion models and proposed a method to exploit this pattern to ignore the computation of attention between many query/key pairs and hence speed up the generation.

**Strengths:**

- The finding (attention tile) is quite interesting and could be useful for future research in the community in this area.
 - the proposed layer-wise optimal search for sparse attention masks is somewhere novel in the paper.

**Weaknesses:**

- This work over-claimed the contribution of the proposed method. Actually, the efficiency improvement is mostly coming from MLCD, which is proposed by another work. The real improvement from the main finding or the proposed 'new' method in this paper is much less than MLCD.
 - Experiment is not thorough. This paper only experimented their method on one DiT based text-to-video generation model.
 - Comparison with other methods is missing. This paper only compared the results from different hyper parameters of the proposed method. Many existing methods that accelerate diffusion models are missing in the paper.
 - The larger diagonal attention is not something new or surprising as the each query token is computing the `correlation` with itself.

**Questions:**

please refer to the weakness section

---

> ### Author Response · Authors · 2024-11-21
>
> We thank the reviewer for acknowledging the novelty of our Attention Tile finding and layer-wise optimal search approach. We summarize the questions here and respond to them individually:
> ## Q1: The main speedup comes from the existing MLCD method rather than the paper's novel contribution (sparse attention)
> A1: We would like to respectfully clarify that our method achieves 2.83x speedup in the attention module (Table 1) and 1.77x end-to-end speedup (listed below), with a sparsity level of 1:7. Our key point here is that although the sparse attention alone does not achieve higher speedup than the MLCD method alone, this speedup is arguably substantial, and is complementary to MLCD. In 3D-DiT, redundancy stems from repetitive diffusion sampling and 3D attention; MLCD addresses the former, while attention distillation targets the latter. The value of our method is that it harmoniously connects them to effectively eliminate 3D-DiT redundancy.
> | Model | Final Score ↑ | Aesthetic Quality | Motion Smoothness | CD-FVD ↓ | Speedup |
> |--------|--------------|------------------|-------------------|-----------|----------|
> | Base | 76.12% | 58.34% | 99.43% | 172.64 | 1.00× |
> | $\text{Base}_{4:4}$ | 76.57% | 58.64% | 99.38% | 171.62 | 1.16× |
> | $\text{Base}_{3:5}$ | 75.53% | 55.47% | 99.01% | 197.35 | 1.26× |
> | $\text{Base}_{2:6}$ | 76.33% | 57.14% | 99.06% | 201.61 | 1.45× |
> | $\text{Base}_{1:7}$ | 77.15% | 57.53% | 98.67% | 322.28 | 1.77× |
>
> ## Q2: Only experimented their method on one DiT based text-to-video generation model.
> A2: We have updated ablation of the attention distill experiment on new model CogvideoX in new thread.
>
> ## Q3: Lacks comparison with other acceleration methods, only showing self-comparisons with different parameter settings
> A3: We thank the reviewer for the suggestion. We included one state-of-the-art work, PAB[1], into our discussion. PAB, developed on spatial-temporal DiTs, reuses attention computation from previous denoising steps to speed up inference. In their best-performing setup, they compute spatial attention, temporal attention, and cross attention every 2, 4, and 6 steps, respectively, which results in an average speedup of 4 in attention. This would translate to less than 4x speedup in an end-to-end setting. As shown in Table 5, our method, combined with consistency distillation, achieves 6.60x-7.80x end-to-end speedup.
> Second, we'd like to point out that PAB falls into the category of methods that leverage the repetitiveness between diffusion steps. Intuitively, it would be less compatible with consistency distillation (which is a well-established method to reduce repeated diffusion steps), because consistency distillation already reduces the repetitiveness between diffusion steps. In contrast, our proposed approach discovers and addresses a new type of redundancy in video diffusion, and shows ample evidence that it is complementary to consistency distillation.
>
> | Model | VBench Performance(%) | Speedup |
> |:---:|:---:|:---:|
> | $\text{PAB}_{\text{246}}$ | -0.09 | <4x |
> | $\text{PAB}_{\text{357}}$ | -2.85 | <5x |
> | $\text{PAB}_{\text{579}}$ | -8.58 | <7x |
> | $\text{Ours}_{r\text{=0.025}}$ | +0.02 | 5.85x |
> | $\text{Ours}_{r\text{=0.050}}$ | -0.11 | 6.60x |
> | $\text{Ours}_{r\text{=0.100}}$ | -0.12 | 7.05x |
> | $\text{Ours}_{r\text{=0.400}}$ | -0.84 | 7.80x |
>
> [1] Zhao, X., et al. (2024). Real-Time Video Generation with Pyramid Attention Broadcast. arXiv preprint arXiv:2408.12588.
>
> ## Q4: The diagonal attention pattern emphasized in the paper is an obvious phenomenon due to the basic property of self-attention
> A4: While the high main diagonal values may be obvious to some audiences, this is only one part of the attention tile phenomenon. The attention tile phenomenon comprises four observations (Figure 1): repetitiveness, large diagonals, locality, and data independence. Such observations have not been revealed by other literature to the best of our knowledge, and form the basis of our methodology.

---

> ### Author Response · Authors · 2024-11-28
>
> ## Q2: Only experimented their method on one DiT based text-to-video generation model.
> A2: We appreciate the reviewer's feedback. We have added comprehensive experiments on the CogVideoX-5B model in Appendix D.2 to demonstrate our method's generalization capability. CogVideoX is based on the MM-DiT architecture, where its attention module concatenates text tokens with video tokens, which differs from Open-Sora-Plan's cross attention module. These experiments demonstrate that our method works effectively on both MM-DiT and Open-Sora-Plan's cross attention architectures.
>
> 1. **Kernel Performance** : We analyze the computation time for a single sparse attention kernel below. The results show that as sparsity increases, computation time decreases significantly. For instance, with a 2:11 attention mask, the execution time reduces to 15.16ms, achieving a 1.72× speedup compared to the full mask.
>
> | Mask | Sparsity (%) | Time(ms) | Speedup |
> |------|--------------|----------|----------|
> | full | 0.00 | 26.03 | 1.00$\times$  |
> | 1 | 14.50 | 24.12 | 1.08$\times$  |
> | 2 | 29.29 | 23.68 | 1.10$\times$  |
> | 3 | 38.30 | 20.51 | 1.27$\times$  |
> | 4 | 48.66 | 17.77 | 1.47$\times$  |
> | 6 | 60.15 | 14.08 | 1.85$\times$  |
> | 12 | 74.11 | 9.99 | 2.60$\times$  |
>
> 2. **Evaluate our method on VBench**
>
> **Experiment Setting**: CogVideoX-5B is profiled using Algorithm 1. For training, the model is trained for 10,000 steps, equivalent to 10 epochs of the dataset. The learning rate is set to 1e-7, and the gradient accumulation step is 1. The diffusion scale factor $\lambda$ is set to 1.
>
> **Quantitative results**: The VBench evaluation results of the knowledge distillation model are shown below. Our model's results are within 1% of the final score with no noticeable drop in several key dimensions. It achieves comparable performance to the original model.
>
> | Model | Final Score | Aesthetic Quality | Motion Smoothness | Temporal Flickering | Subject Consistency | Overall Consistency | Speedup |
> |-------|--------------|-------------------|-------------------|---------------------|---------------------|-------------------|----------|
> | Base | 77.91% | 57.91% | 97.83% | 97.34% | 92.27% | 26.13% | 1.00$\times$  |
> | $\text{Ours}_{r\text{=5}}$ | 77.15% | 51.18% | 96.67% | 97.18% | 90.89% | 26.02% | 1.34$\times$  |
>
> **Qualitative results**: In Appendix D.2. Figure 7, we demonstrate that our method shows robust performance in processing dynamic, complex scenes while maintaining high-quality video output using the prompt from CogVideoX official website.

---

### Official Review · Reviewer_AbNB · 2024-11-02

**Soundness:** 3
**Presentation:** 3
**Contribution:** 3
**Rating:** 6
**Confidence:** 4

**Summary:**

The paper tackles the inefficiency of DiTs used in video diffusion model. The speedup of the presented method comes from two sources: 1) pruning the large full 3D attention of VDM DiTs and 2) distilling the model into a multi-step consistency model.
The authors identify a repetitive tile-like pattern, termed "Attention Tile," in the 3D attention maps of video data. Leveraging this pattern, they propose a new family of sparse 3D attention mechanisms that reduce the computational complexity from quadratic to linear with respect to the number of video frames.
To further accelerate the inference process, the paper introduces a multi-step consistency distillation (MCD) technique. By dividing the sampling trajectory into segments and performing consistency distillation within each, the number of sampling steps required for video generation is significantly reduced.
Results show that the method achieves good speedup without suffer much performance, using limited training data.

**Strengths:**

1. The paper makes a significant contribution by discovering the "Attention Tile" phenomenon in 3D full attention Diffusion Transformers (DiTs) for video data. This insight into the redundancy and repetitive patterns within attention maps is a valuable addition to the understanding of how attention mechanisms function in video generation models.
2. Building on the Attention Tile observation, the authors propose a new family of sparse 3D attention mechanisms that reduce computational complexity from quadratic to linear concerning the number of video frames. This is a substantial improvement that directly addresses the inefficiency issues in existing models.
3. The introduction of the EFFICIENT-VDIT framework is a well-thought-out approach that combines multi-step consistency distillation, layer-wise sparse attention mask searching, and knowledge distillation. This pipeline effectively accelerates inference while maintaining high metrics.
4. Achieving these results using only 0.1% of the pretraining data is notable. It indicates that the method is not only computationally efficient but also data-efficient, which is advantageous when large datasets are not readily available.

**Weaknesses:**

1. The paper could benefit from a more in-depth discussion of the trade-offs involved, such as the balance between sparsity level and video quality or the impact on different types of video content (e.g., fast-moving vs. static scenes). For instance, why don't you directly use the demo videos on OpenSORA's websites and compare the qualitative results? They provided both static scenes with only relative camera poses and more dynamic scenes, e.g. filming of an explosion scene.
2. The method relies on the observation that the Attention Tile pattern is data-independent. If this assumption does not hold for certain types of video data (e.g., highly dynamic scenes), the efficiency gains might not translate, potentially limiting the method's applicability.
3. The use of only 0.1% of the pretraining data raises concerns about the generalization capabilities of the accelerated model. While performance loss is minimal on tested datasets, the model may underperform on unseen data or less common video scenarios.
4. While the paper uses VBench and FVD for evaluation, these metrics may not capture all aspects of video quality, such as temporal coherence in more complex scenes or perceptual quality under different conditions. Including additional metrics or user studies could provide a more comprehensive assessment. This is especially concerning combined with weakness #2, since FVD is commonly known as a weak metric that focuses strongly on independent frames rather than overall video coherence. Overall, the evaluation seems to favor more static videos rather than highly dynamic videos, and I suspect the attention pruning would encourage such results too. A metric that takes motion into account is Content-Debiased FVD [1], but ideally, this is more suitable via a user study (even though I do not think this is necessary for the rebuttal stage, but better prepare it for another iteration of the paper).
5. Inherit my point in #2 and #4, the paper does not provide any video data, making it challenging to assess the actual quality of the generated contents. From my point of view, a VDM paper should always be accompanied with as many videos as possible within the supplemental material size limit. Again, a good set would be the demo videos on OpenSORA's websites. They provided a wide range of descriptions and all the corresponding text prompts --- supposedly those prompts would work well on OpenSORA.

[1] Ge et al., On the Content Bias in Fréchet Video Distance, in CVPR 2024.

**Questions:**

N/A

---

> ### Author Response · Authors · 2024-11-21
>
> We thank the reviewer for the positive feedback on the sparse attention mechanisms in the video model and distillation framework. We summarize the questions here and respond to them individually:
>
> ## Q1: Tradeoff analysis: Lacks detailed discussion of trade-offs between sparsity and video quality, especially for different types of content (static vs. dynamic scenes)
> A1: From Table 5, we can observe that when sparsity increases, VBench dimensions such as total score and motion smoothness decrease, while higher sparsity leads to better approximation of the original image. Also, as shown in Table 1, as sparsity increases, the sparse kernel requires less execution time, thus reducing inference time. To address this trade-off, we present an optimization method based on the Lagrangian method in Appendix A, which considers how to achieve higher sparsity under given acceleration ratio constraints, thereby better approximating the original image. We have included updated video examples from VBench and examples similar to those on OpenSora's website featuring dynamic scenes in Appendix E.
>
> ## Q2: Concerns about model generalization when using only 0.1% of pre-training data.
> A2: We thankfully agree with the reviewer that more data would improve model generalization ability. In fact, we show that our method has already had good generalization ability with only 0.1% of pre-training data on VBench.Note that VBench already contains a diverse suite of test prompts, which suffices as a proof-of-concept. We believe distillation with more data will further boost the performance, but defer it as a future work due to the large amount of GPU hours required.
>
> ## Q3: Current metrics (VBench and FVD) may not fully capture video quality, especially temporal coherence -> Content-Debiased FVD.
> A3: Following your suggestion, we have updated our evaluation to include Content-Debiased FVD scores, which more effectively capture the temporal coherence of video quality. This additional metric provides better insight toward per-frame quality over temporal realism and helps identify its sources. As shown in the table below, the base model achieves better performance on Content-Debiased FVD, which aligns with our expectations.
>
> | Model | FVD ↓ | Content-Debiased FVD ↓ |
> |:---:|:---:|:---:|
> | Base| 381.1 | 172.6 |
> | MLCD | 438.1 | 190.5 |
> | $\text{Ours}_{r\text{=0.025}}$ | 351.6 | 186.8 |
> | $\text{Ours}_{r\text{=0.050}}$ | 357.4 | 195.6 |
> | $\text{Ours}_{r\text{=0.100}}$ | 345.6 | 204.1 |
> | $\text{Ours}_{r\text{=0.200}}$ | 356.9 | 223.8 |
> | $\text{Ours}_{r\text{=0.400}}$ | 380.2 | 231.7 |
>
> ## Q4: Lack of sufficient video samples in supplementary materials.
> A4: We have included updated video examples from VBench and examples similar to those on OpenSora's website featuring dynamic scenes in Appendix E, demonstrating our capability to handle rapid, large-scale motions.

---

### Official Review · Reviewer_tMR2 · 2024-11-04

**Soundness:** 3
**Presentation:** 3
**Contribution:** 2
**Rating:** 6
**Confidence:** 3

**Summary:**

This paper addresses the acceleration of 3D full attention video generation models, focusing on sparsifying 3D attention and reducing sampling steps. The authors propose an algorithm for searching optimal sparse attention masks based on the observed Attention Tile phenomenon and combine this with a consistency distillation method to reduce the number of steps, resulting in an accelerated version of DiT while striving to maintain generation quality.

**Strengths:**

The paper effectively optimizes DiT using sparse attention and MCD, and the proposed framework demonstrates commendable speed results alongside assured generation quality. Specific strengths include:

- The identification of the Attention Tile phenomenon, accompanied by a detailed analysis, provides a background for the design of sparse attention masks and proposes an algorithm for searching optimal mask sets. Comprehensive evaluation experiments validate the effectiveness of this method.
- The integration of the consistency distillation method leads to a complete acceleration framework, with rigorous ablation studies confirming the framework's soundness and ensuring generation quality. The FVD metric significantly outperforms the use of the MLCD method alone.

**Weaknesses:**

While the paper is rich in content, there are still potential issues to consider: According to Table 7, the acceleration benefits from sparse attention masks are not substantial, with noticeable quality degradation occurring beyond a 1.45× acceleration. Although there is some improvement when combined with MLCD (compared to a 5× acceleration), the effectiveness of the design based on the Attention Tile, which is a core contribution of the paper, appears insufficient here.

**Questions:**

- The changes in Dynamic Degree seem to exhibit a certain trend; are there any related experimental analyses available?
- There is a discrepancy between the acceleration results in Table 1 and Table 7. Could you please provide the specific experimental parameter differences (as they seem to be absent in the paper)?

---

> ### Author Response · Authors · 2024-11-21
>
> We thank the reviewer for the positive feedback on the Attention Tile phenomenon analysis and distillation framework. We summarize the weaknesses and questions here and respond to them individually:
>
> ## Q1: Attention distill improves little compared to MLCD
> A1: Attention distillation is a method that reduces redundancy that MLCD cannot address and is orthogonal to it. The redundancy in 3D-DiT comes from two sources: the repetitive sampling of diffusion and the redundancy of 3D attention. MLCD can only address the first part, while the goal of attention distillation is to address the second part. In Tables 1 and 2, we show that attention distillation itself provides substantial speedup. In our method, these two components can be combined to jointly eliminate redundancy in 3D-DiT inference.
>
> ## Q2: Dynamic Degree trend analysis
> A2: As discussed in the VBench paper, models often exhibit trade-offs between temporal consistency and dynamic degree metrics. Quantitatively, we measure our models and variants across the overall VBench dimensions to justify the quality of our method. Qualitatively, we provide more samples in Appendix E from the VBench prompt to demonstrate that our method improves motion dynamics while maintaining acceptable consistency. Overall, with higher sparsity, our method captures richer motions between frames but with a trade-off in temporal consistency.
>
>
> ## Q3: Discrepancy between Table 1 and Table 7 results
> A3: The difference between Tables 1 and 7 comes from different measurement scopes. Table 1 measures only the attention operation speedup, while Table 7 shows the speedup of the entire model (including overhead from normalization and MLP layers). For example, with the 1:7 ratio, while attention alone achieves a 2.83x speedup, the full model only achieves a 1.77x speedup due to other components.

---

### Author Response · Authors · 2024-11-21
**General response**

We thank all reviewers for their positive feedback. We are encouraged that they all appreciated our paper for the following reasons:
- Discovery and analysis of the Attention Tile phenomenon with comprehensive observations beyond just diagonal patterns.
- Novel sparse attention mechanism achieving significant speedup complementary to existing MLCD methods.
- Layer-wise optimal search for sparse attention masks demonstrating strong generalization.

Based on the feedback, we mainly update the manuscript with:
- More comprehensive video samples in Appendix E.
- Added Content-Debiased FVD metrics for better video quality evaluation.
- Expanded ablation studies demonstrating the separate effects of sparse attention and MLCD in Appendix D1.
- Experiments on additional state-of-the-art video diffusion architectures CogVideoX in Appenedix D2.

---

### Meta-Review · Area_Chair_2P1k · 2024-12-20

**Metareview:**

The paper is concerned about the task of accelerating the 3D attention in video generators. The reviewers ranked the paper as a borderline, acknowledging the strengths, but also listing a number of weaknesses. They highlight the attention tile idea proposed in the paper. The paper reports favorable VBench scores. At the same time, the AC was surprised to find that the paper reports no video examples in the supplement. This is very uncommon for video papers, in fact, and here AC agrees with Reviewer AbNB and tExx, that the authors should include as many samples as they can. Currently only several frames are concatenated and pasted to the paper. In this form, it's not really possible to analyze the quality of the videos and understand the value of the method. VBench scores are not sufficient alone. There can be temporal artifacts, flickering, inconsistencies that it's not possible to detect when looking at frames. After looking at the examples, the AC believes that the visual quality drops (check Fig 11 the dog example), also there are no prompts given.

The AC went through the discussion, the paper, and the provided video examples. The authors also shared a message with AC. The AC believes that the provided examples are not sufficient for the video paper. None of the reviewers champions the paper strongly.

The AC would like to encourage the authors to update the manuscript with the details they provided during the discussion period (including hundreds of video examples and comparisons, not frames!) and submit to a future venue.

**Additional Comments On Reviewer Discussion:**

There was a somewhat reasonable exchange of messages between the authors and reviewers. Some concerns were solves, others remained.

---

### Decision · Program_Chairs · 2025-01-22

Reject